# Can Vision Language Models Learn Intuitive Physics from Interaction?

Luca M. Schulze Buschoff [1]  Konstantinos Voudouris [1]  Can Demircan [1]  Eric Schulz [1]

## Abstract

Pre-trained vision language models do not have good intuitions about the physical world. Recent work has shown that supervised fine-tuning can improve model performance on simple physical tasks. However, fine-tuned models do not appear to learn robust physical rules that can generalize to new contexts. Based on research in cognitive science, we hypothesize that models need to interact with an environment to properly learn its physical dynamics. We train models that learn through interaction with a simulated environment using reinforcement learning. While learning from interaction allows models to improve their within-task performance, it fails to produce models with generalizable physical intuitions. We find that models trained on one task do not reliably generalize to related tasks, even if the tasks share visual statistics and physical principles, and regardless of whether the models are trained through interaction.

## 1. Introduction

A central goal of machine learning research is to build machines that think and behave like people do. Lake et al. (2017) propose that human-like machine learning models must be capable of reasoning about their environment and its physical, social, and causal structure. These capabilities are often referred to as intuitive theories (Baillargeon et al., 1995; Spelke, 1990; Spelke & Kinzler, 2007). Here, we focus on *intuitive physics* — the fast, implicit ability to understand and predict the physical properties and interactions of objects (Battaglia et al., 2012; Piloto et al., 2022). Its canonical components include object permanence, continuity, solidity, and support (Margoni et al., 2024; Chiandetti & Vallortigara, 2011; Wood et al., 2024). Our work studies these intuitions in VLMs (Battaglia et al., 2013; Lake et al., 2017; Piloto et al., 2022; Schulze Buschoff et al., 2025a;b).

[1]Institute for Human-Centered AI, Helmholtz Munich, Oberschleißheim, Germany. Correspondence to: Luca M. Schulze Buschoff <lucaschulzebuschoff@gmail.com>.

*Proceedings of the $43^{rd}$ International Conference on Machine Learning*, Seoul, South Korea. PMLR 306, 2026. Copyright 2026 by the author(s).

Recent work has established that vision language models (VLMs), models that receive visual and textual inputs, are still limited in their understanding of the physical world and its causal structure (Jin et al., 2023; Balazadeh et al., 2025). VLMs do not perform well on standard visual cognition tasks — such as tasks testing intuitive physics — and they do not show a good fit with human behavior (Schulze Buschoff et al., 2025a). While supervised fine-tuning (SFT) can make models perform well on the fine-tuning task, fine-tuned models do not appear to learn generalizable intuitions about the physical world (Schulze Buschoff et al., 2025b).

A prominent idea in cognitive science is that humans learn a robust understanding of their world by interacting with it (Gibson, 1979; Merleau-Ponty, 1945; Varela et al., 1991). The key claim is that humans learn robust, generalizable concepts for explaining and predicting their world not merely from passive observation and symbolic abstraction, but from actively interacting with their environment's dynamics (Barsalou, 1999; Clark, 1998). Some have argued that directly experimenting with the physical properties of objects in the environment allows children to test their hypotheses about their environment (Gopnik et al., 1999). In contrast to passively observing the interactions of other people with an environment, they learn much more from trying, and often failing, to predict how the environment will evolve given their own actions (Smith, 1982; Chu & Schulz, 2020; Nicolopoulou, 1993; Smith & Gasser, 2005; Schulz & Bonawitz, 2007). While the important role of interaction is slowly being recognized in generative model training (Silver & Sutton, 2025; Motamed et al., 2026), its merit for teaching vision language models visual cognitive abilities such as intuitive physics has not yet been explored.

In this paper, we present a first attempt at evaluating the role of interaction for learning intuitive physics in VLMs. Interaction can be operationalized in several ways (Shapiro & Spaulding, 2021), from one- and multi-step reinforcement learning (RL) to multi-sensory robotics. We operationalize interaction in the context of one-step RL, defining an *environment*, *action space*, and *reward function* (Sutton et al., 1998). VLMs are presented with an image of a stack of colored blocks generated by a physics engine. They must for example respond with an action sequence to move another block to build a taller, stable tower, receiving a reward that depends on the stability of the resulting tower.

We compare models that are trained to build towers through trial-and-error (the *interactive* condition) with models that are shown examples of optimal action sequences to build stable towers (the *non-interactive* condition). Similarly to how children appear to learn generalizable physical intuitions by playing with objects (Piaget, 1952), we propose that learning to build towers through interaction with the physics of the environment will enable VLMs to learn those same intuitions. Following this line of argument, we hypothesize the following:

1. Models in the interactive condition will generalize better to building new towers not seen in their training data, compared to the non-interactive condition.

2. Models in the interactive condition will generalize better to a new task, such as judging the stability of a tower, compared to the non-interactive condition.

We test these hypotheses mainly by evaluating the textual outputs of VLMs. However, it is possible that models might have the knowledge required to solve the task, but cannot produce textual outputs in the right format. We explore this distinction between model *competence* and model *performance* (Chomsky, 1965) by decoding model activations layer-wise to see how predictive they are of key physical quantities. We thus further hypothesize that these quantities will be more decodable at later model layers in models trained in the interactive condition compared to the non-interactive condition.

We find no noticeable differences between the interactive and non-interactive conditions, both in and outside of the training tasks. Both methods yield models that perform at ceiling on the tasks they are trained on, but neither method produces models that reliably generalize to new physical tasks. While we find that physical quantities like tower stability are highly decodable from model activations, neither post-training method successfully converts this competence into reliable performance on new tasks.

## 2. Related Work

Prior work has documented broad failures of VLMs on physical reasoning benchmarks (Ballout et al., 2025; Han et al., 2025). Despite recent advances in architectures and training methods, VLMs continue to struggle on simple visual tasks that are trivial for any human observer, such as counting objects in a scene or making judgements about their interactions (Rahmanzadehgervi et al., 2024; Schulze Buschoff et al., 2025a; Balazadeh et al., 2025). Campbell et al. (2024) suggest that these failures originate from models' struggling with the binding problem. Earlier studies showed that pre-trained VLMs struggle to attend to multiple objects at the same time (Frankland et al., 2021), however newer models

have narrowed the gap to human baselines for multi-object discrimination (Chen et al., 2024; Feng et al., 2025; Izadi et al., 2026; Jia et al., 2024) and counting (Guo et al., 2025; Jiang et al., 2024; Qharabagh et al., 2024; Vo et al., 2025).

Supervised fine-tuning has emerged as an efficient way to overcome model limitations through extensive post-training on specific problems (Han et al., 2024). These methods have also proven useful for aligning models towards more human-like outputs (Binz et al., 2024; Hussain et al., 2024). However, SFT with VLMs appears to have a limited effect on their ability to learn generalizable physical intuitions (Schulze Buschoff et al., 2025b) and interact reliably with physical environments (Mecattaf et al., 2024). One plausible hypothesis is that SFT simply allows VLMs to learn useful shortcuts for specific tasks (Geirhos et al., 2020). Similarly, Motamed et al. (2026) argue that large generative video models are likely making predictions about the physical world without a proper understanding of its underlying physics. They suggest that a lack of active interaction with the physical world could be the limiting factor. Our study therefore seeks to explore whether models' understanding of physics can be improved through active interaction with an environment.

In line with this proposal, online reinforcement learning, a paradigm in which models learn through interaction with an environment, has been argued to generalize more robustly than SFT (Chu et al., 2025), which is akin to behavioral cloning (Wu et al., 2025). Online RL refers to updating the model sequentially based on actions it has taken in an environment, whereas offline RL refers to updating the model based on a fixed set of state-action pairs collected using another policy (Levine et al., 2020; Ostrovski et al., 2021). Chu et al. (2025) train VLMs on arithmetic reasoning and simple navigation tasks and find that online RL trained models generalize better than models trained with SFT. In contrast, our work focuses on established intuitive physics tasks in cognitive science: building and judging the stability of block towers.

Closest to our setup, recent work on spatial reasoning using stacked-object stability tasks shows that SFT and RL fine-tuning on Qwen2.5-VL can improve task performance (Han et al., 2025) — but this study does not systematically test generalisation between different physical tasks. Work on world modelling and understanding of physical transformations in VLMs suggests that current models lack robust internal representations of physical phenomena like conservation and causality (Gao et al., 2025; Luo et al., 2026). However, probing analyses of VLM representations show that vision encoders capture physical plausibility cues even when behavioural performance fails (Ballout et al., 2025), motivating our decodability analysis.

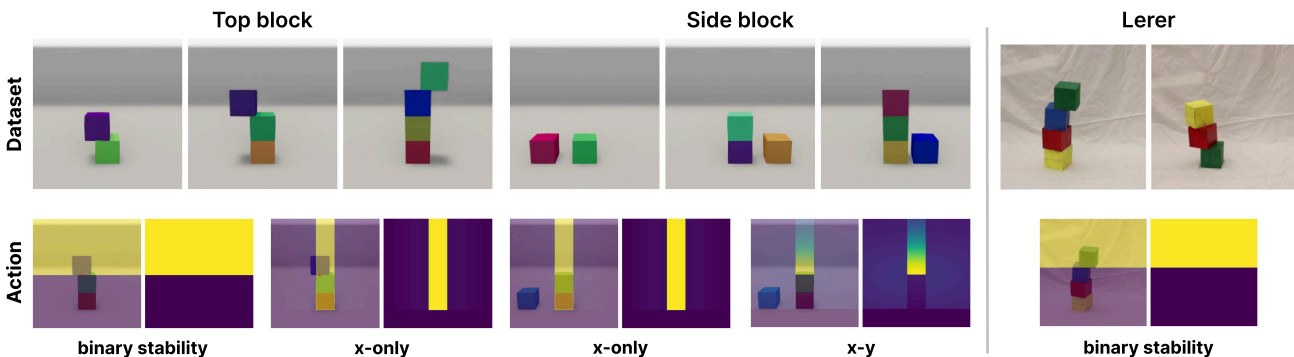

*Figure 1.* Overview of all combinations of datasets and action types. **Datasets:** We train models on two related datasets, one where the block on top of a tower is displaced, and one where a block is displaced on the ground next to a tower. **Actions:** For each dataset, we train models on two action types. *Binary stability* requires models to make a binary judgment on whether a given tower is stable. For *x-only*, models need to give a single value by which the displaced block should be moved to build a more stable or bigger tower. The *x-y* task is the same as *x-only*, but with an added dimension — here the block needs to be moved to the side and up. The heatmaps visualize the reward functions for all combinations of dataset and action (see Section A.3 for more details). We train models on all four combinations of dataset and action types using GRPO to test whether models trained through interaction with an environment learn generalizable physical intuitions. We also evaluate models on an external dataset of real wooden block towers, taken from Lerer et al. (2016).

## 3. Methods

### 3.1. Datasets

We construct two tower block datasets for our experiments, each consisting of stacks of 2-4 randomly colored cubes. 256×256 pixel RGB images are taken from a fixed camera angle in the ThreeDWorld environment (Gan et al., 2020). We keep the camera angle and block sizes fixed throughout, so that the models are able to learn the mapping between pixel space and ground truth distance. Both datasets feature towers that consist of perfectly stacked blocks except for one block. In the first dataset, **top block**, this block is on top of the tower but displaced to the left or the right (see Fig. 1 and Fig. 9 in the Appendix).

In the second dataset, **side block**, the block is on the floor next to the tower, also either to the left or the right (see Fig. 1 and Fig. 10 in the Appendix). As an additional evaluation dataset, we also use an independent dataset of real images of tower blocks from Lerer et al. (2016; see Appendix A.1).

### 3.2. Tasks

Using these datasets, we construct four tasks. Using the top block and Lerer et al. (2016) datasets, the **binary stability** task requires the model to give a judgment on whether a given tower is stable or not. In contrast, the **x-only** task requires the model to return a single integer that moves the block along the $x$-axis (see Fig. 1). Here, the goal is to improve the stability of the tower by moving the block closer to the centre. In both tasks, the model must attend to the displacement of the top block from the centre point of the tower, both to judge its stability and to choose an appropriate counter-displacement to stabilize the tower. However, the latter case is framed interactively.

The **x-only** task for the side block dataset again requires the model to give an integer to move the block to the most central position. The x-only tasks are identical except for the range of correct integers due to the different block displacements. The **x-y** task requires the model give two integers to move the block in both the $x$- and $y$-dimensions (see Fig. 1). Moreover, models should be readily able to generalize from the x-y task to the x-only tasks, due to their being identical problems on the $x$-dimension. The prompts for each task are included in Appendix A.5.

### 3.3. Fine-Tuning Methods

We fine-tune the 8B parameter 4-bit quantized version of the Qwen3-VL model (Yang et al., 2025) using the unsloth library (Han et al., 2023). We employ Parameter Efficient Fine-Tuning (PEFT; Han et al., 2024) — rather than updating all model weights we update small low-rank adapters inserted layer-wise in the model (QLoRA; Dettmers et al., 2024; Hu et al., 2021).

We are interested in training models that learn from interaction. To implement this, we train models using RL with Group-Relative Policy Optimization (GRPO). As a non-interactive baseline, we compare the GRPO models to models post-trained with SFT. We outline each method in turn. Note that in Section 4.5 and Section A.8 in the Appendix, we also describe results from experiments with other models and a different RL algorithm.

Models are trained on 10.000 images for each respective combination of dataset and task. Models receive a single image per step. The GRPO post-trained models generate 16 responses for each sample, over which the reward is calculated. Models are evaluated on a separate set of 10.000 images for each task.

**Group-Relative Policy Optimization**   In the reinforcement learning setting, the set of all model and adapter weights ($\theta$) is considered the policy, $\pi_\theta$. It takes the text prompt and image as input (observations of the state of the environment), and produces a token sequence as actions. For a batch of $M <$ prompt, image $>$ pairs, $\{p_1, ..., p_M\}$, the model produces a set of $M \times N$ completions $\{c_{1,1}, ..., c_{1,N}, ..., c_{M,N}\}$. These completions are rewarded using a reward function, giving a set of rewards $\{r_{1,1}, ..., r_{1,N}, ..., r_{M,N}\}$. We use $N = 16$ in our experiments.

We compute the loss for some prompt $p$ as:

$$\mathcal{L}(\theta) = -\frac{1}{\sum_{i=1}^{N}|c_i|} \sum_{i=1}^{N} \sum_{t=1}^{|c_i|} \Big[ \min\Big(\frac{\pi_\theta(c_{i,t}|q, c_{i,<t})}{\pi_{\theta_{old}}(c_{i,t}|q, c_{i,<t})}\hat{A}_{i,t},$$
$$\text{clip}\Big(\frac{\pi_\theta(c_{i,t}|q, c_{i,<t})}{\pi_{\theta_{old}}(c_{i,t}|q, c_{i,<t})}, 1 \pm \eta\Big)\Big)\hat{A}_{i,t}\Big]$$

Where $|c_i|$ is the length of the completion, in tokens, and $\hat{A}_{i,t}$ is the normalized reward (advantage) for $|c_i|$:

$$\hat{A}_{i,t} = \frac{r_i - \text{mean}(\{r_1, ..., r_n\})}{\text{std}(\{r_1, ..., r_n\})}$$

Following common practice (Hu et al., 2025; Liu et al., 2025; Yu et al., 2025), we exclude the original KL-divergence term used in (Shao et al., 2024). We update the adapter weights with gradient ascent over $\mathcal{L}(\theta)$.

**Supervised Fine-Tuning**   Using a labelled dataset, model weights are updated using batch gradient descent over the token-level cross-entropy loss:

$$\mathcal{L}(\theta) = -\sum_{t=1}^{T} \log p_\theta(y_t|y_{<t})$$

where $\theta$ is the set of model and adapter weights, $T$ is the ordered set of target completion tokens given a prompt, $y_t$ is the target token at step $t$, and $y_{<t}$ is the set of ordered target completion tokens prior to $t$. Only the adapter weight subset of $\theta$ are actually updated.

**PEFT Hyperparameters**   We keep the hyperparameters across both fine-tuning methods as consistent as possible. All adapters are the same size and injected in all layers of the model. Specifically, we inject a matrix $W_a$ at each layer, which is the product of two low-rank matrices, $M_1 \in \mathbb{R}^{d \times r}, M_2 \in \mathbb{R}^{r \times k}$, where $d, k$ are the input and output dimensionalities respectively, and $r << d, k$. During the forward pass, the outputs of the full weight matrix for that layer are summed with the outputs of the adapter, subject to some scaling factor $\frac{r}{\alpha}$. We use $r = \alpha = 16$ everywhere.

We use stochastic gradient descent with the Adam optimizer. We train all models for 10,000 steps on single 80GB A100 GPUs.

## 3.4. Reward Functions

For the binary stability task, where models have to give a binary response judging the stability of a given block tower, we use three distinct reward values: $-1$ for non-parseable answers, $0$ for legal but incorrect answers, and $1$ for legal and correct answers.

For the x-only task where models reply with a single integer, we set the reward to $-5$ for non-parseable completions. For answers that are parsed correctly we use two different Gaussian functions based on the distance to the center. As answers get closer to the center, they are rewarded more. For answers that result in an unstable tower, we calculate a weaker function as $2 \cdot e^{(-d^2)} - 2$, where $d$ is the distance on $x$ from the optimal position. For answers that result in a stable tower, we compute the reward as $20 \cdot e^{(-d^2)}$ (see Fig. 1 and Fig. 9 in the Appendix for a visualization).

For the *x-y* task where models must reply with two integers, we again set the reward to $-5$ for non-parseable answers. For answers that move the block below the floor, we set the reward to $-4$. For all other parseable answers, we again compute Gaussian reward functions depending on the euclidean distance between the final position of the moved block and the optimal position on top of the tower. For answers that are above ground but do not result in a stable bigger tower, we calculate the reward as $2 \cdot e^{(-d^2)} - 2$. For answers that are within the tower, we compute $2 \cdot e^{(-d^2)} - 4$. And for answers that result in a stable bigger tower, we compute the reward as $20 \cdot e^{(-d^2)}$ (see Fig. 1 and Fig. 10 in the Appendix for a visualization).

## 4. Results

We evaluate performance on held-out instances from the post-training task (4.1), generalization to the other tasks (4.2), and generalization to the binary stability task with real images of block towers (4.3).

### 4.1. Post-Training Performance Improvement

GRPO improves performance of the pre-trained model on all post-training tasks (see the diagonal in Fig. 2). On the binary stability top block task, where models have to give a binary judgment on the stability of a block tower, the GRPO model achieves a mean test accuracy of 0.943 after 10,000 steps (the ceiling here is 1, for all other tasks it is 20). Similarly, the SFT model trained without interaction achieves a mean test accuracy of 0.969.

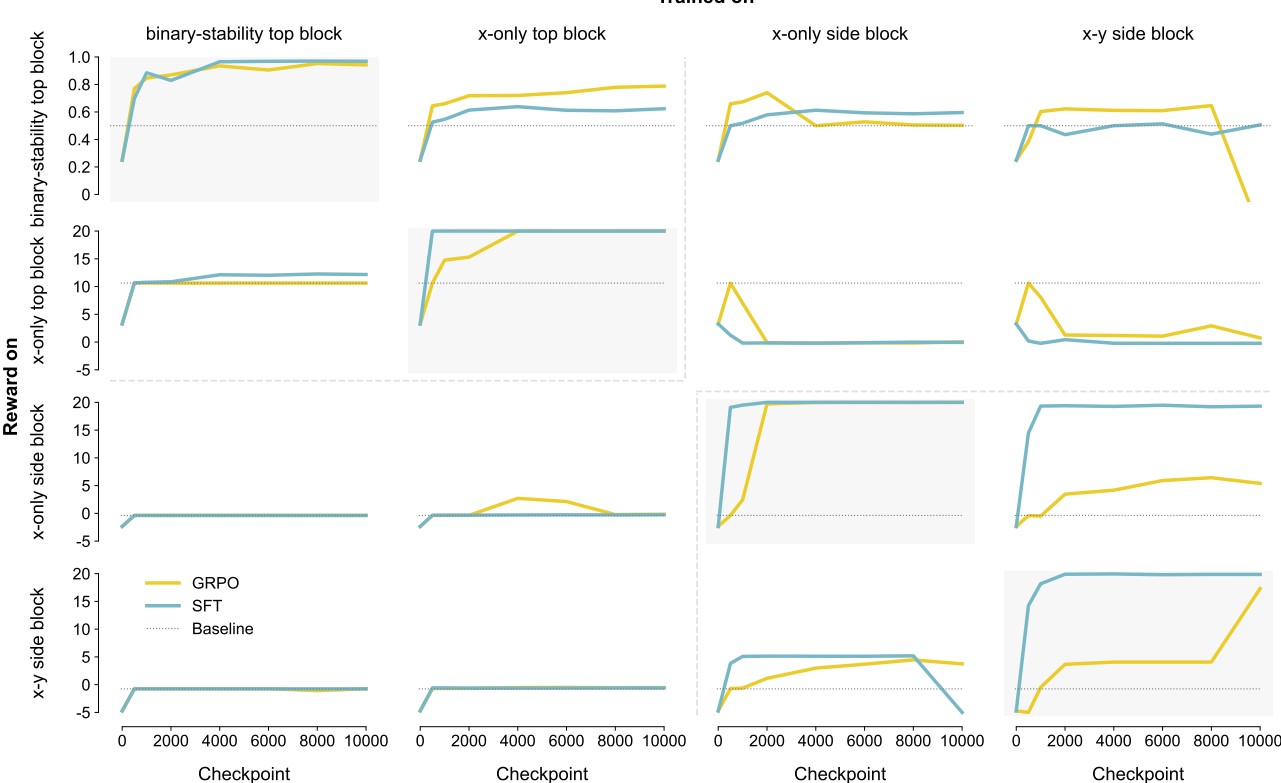

*Figure 2.* Performance by test task and training task for Qwen3-VL-8B. Rows show models evaluated on a given task. Columns show models trained on a given task. The blue and orange lines show the performance of the models trained with SFT and GRPO, respectively. The grey dotted line shows the baseline for the evaluation task. Plots on the diagonal show within-task performance, meaning models are evaluated on the same task they are trained on. All other subplots represent some degree of generalization.

On the x-only top block task, models are asked to return a single integer to move the top block into a more stable position. Here, the GRPO model achieves a mean test reward of 19.999 after 10,000 steps. The SFT model achieves the same score. For x-only on side block, models are also asked to return an integer to move a block, here one that is misplaced on the floor to the left or right of the tower, to the center of the image. The GRPO model again achieves a mean test reward of 19.998, the SFT model 20. The x-y task on side block is similar to the x-only task, however models here also have to return a second integer to move the block not only to the left or right but also up into the most stable position on top of the tower. Here, the GRPO model gets a mean test reward of 17.313. In contrast, the SFT model gets a mean test reward of 19.86.

To summarize, we find that training with interaction through GRPO allows models to improve performance on their training task. However, the same is true for the SFT models that are trained without interaction — as such, we find no direct benefit of training with interaction when it comes to improving within task performance.

### 4.2. Generalization to Related Tasks

We are not just interested in models that perform well on a single physical task, but rather models that robustly generalize from their experience to solve new tasks (Collins et al., 2022; Geirhos et al., 2018; Griffiths & Tenenbaum, 2009). Therefore, we test whether models post-trained on single tasks can generalize to new, related tasks. To test this, we evaluate all post-trained models on all tasks.

We find that no model reliably generalizes to all other tasks, even if they are trained with interaction (see Fig. 2). However, we find that there is slight generalization to different tasks on the post-training data (see the the upper left and lower right quadrants of Figure 2). For example, the GRPO model trained on the x-y side block task also performs above the baseline on the x-only side block task, achieving a mean reward of 5.396 (the model overfits to returning two integers, filtering only legal answers yields a mean reward of 12.772). This is to be expected because x-only is a subset of the x-y task which requires only the output of the x variable that the model has learned. The SFT model achieves a mean test reward of 19.316 (see Table 1 for the full set of model results after 10.000 steps).

| Evaluated on | Trained for 10.000 steps with GRPO (SFT) on | | | |
|---|---|---|---|---|
| | binary-stability top block | x-only top block | x-only side block | x-y side block |
| binary-stability top block | 0.943 (0.969) | 0.788 (0.624) | 0.503 (0.596) | -0.264 (0.506) |
| x-only top block | 10.626 (12.170) | 19.999 (20) | 0.042 (-0.06) | 0.75 (-0.214) |
| x-only side block | -0.373 (-0.373) | -0.152 (-0.256) | 19.998 (20) | 5.396 (19.316) |
| x-y side block | -0.723 (-0.759) | -0.535 (-0.582) | 3.738 (-5) | 17.313 (19.86) |

*Table 1.* Results for Qwen3-VL-8B after post-training for 10.000 steps with GRPO (SFT). Rows show models evaluated on a given task. Columns show models trained on a given task. For results over different checkpoint see Fig. 2 above.

We also find a slight carry-over between the binary-stability and x-only tasks on the top block data — solving both tasks requires the same task variable, the x-offset of the top block. In the x-only condition, the model is explicitly forced to learn this variable as the amount the top block has to be moved by in order to put it in the most stable position. It is also the single variable needed to solve whether a given block tower is stable or not. Despite this, we only find very limited generalization between the two conditions, highlighting how constrained generalization from either post-training method is. When evaluated on binary-stability top block, the GRPO models trained for 10.000 steps on x-only top block, x-only side block, and x-y side block get accuracies of 0.624, 0.503, and −0.264 respectively. The SFT models trained on the same conditions perform at 0.624, 0.596, and 0.506 (since this is a binary task the random baseline is 0.5, but illegal answers can pull the reward down — see Appendix Section A.2 for tables with only legal answers).

To conclude, models perform well on their fine-tuning task (see the diagonal in Fig. 2) and they show slight patterns of generalization to other tasks on the same data (from x-only top block to binary stability top block). However, generalization to other data but with the same task is limited (from x-only top block to x-only side block).

### 4.3. Generalization to Real Images

To test whether models learn physical intuitions that can generalize to real images, we test them on 100 images from Lerer et al. (2016). These images show real wooden block towers consisting of 2 to 4 colored wooden blocks, which are either stable or unstable.

When evaluated on this data, the GRPO models trained for 10.000 steps on binary-stability top block, x-only top block, x-only side block, and x-y side block get accuracies of 0.6, 0.57, 0.52 and 0.31 respectively. The SFT models trained on the same conditions perform at 0.59, 0.53, 0.55 and 0.57 (see Appendix Section A.2 for tables with only legal answers).

While we can see some transfer, for example from our binary-stability task to these real images, all models perform below the human average[1] (see Fig. 3). Furthermore, we again find no visible benefit of training models with interaction over supervised training when it comes to generalization.

### 4.4. Decodability Analysis

To further explore whether the models have learned some task-general features, we analyzed activations during the forward pass to see if the models represent the information necessary to generalize to our set of intuitive physics tasks (see Section A.6 in the Appendix for more details). If the activations encode this information, it suggests that the models have the competence to solve the intuitive physics tasks, but fail to convert that into good performance.

We find that the binary stability of a tower is already trivially decodable in the base model, and it does not change in any substantial way through either fine-tuning method (see Fig. 13 in the Appendix). This is likely because there exists an obvious pixel level shortcut where tower stability can be determined from a small set of pixels along the horizontal center line in the image. However, while this information is decodable from everywhere throughout the model, the base model performs the binary stability task at much lower accuracies than achieved by the linear probes.

The same is true for the offset of the top block. Again, already in the base model, the x-offset is highly decodable and differences between the base, GRPO, and SFT models are small. This analysis suggests that while the models have the competence to solve either task, this does not translate to good performance, hinting at shortcut learning. This invites the question of what information the models use when solving these tasks. To better understand this, we look at what they attend to in an image. We compare the attention maps of post-trained models to those of the base model to see if they learned to focus on specific parts of the image. However, the attention maps are noisy and do not reliably show differences in model attention as a result of either post-training method (see Appendix Section A.7).

---

[1]We calculate the human average based on publicly available, anonymized data taken from Schulze Buschoff et al. (2025a)

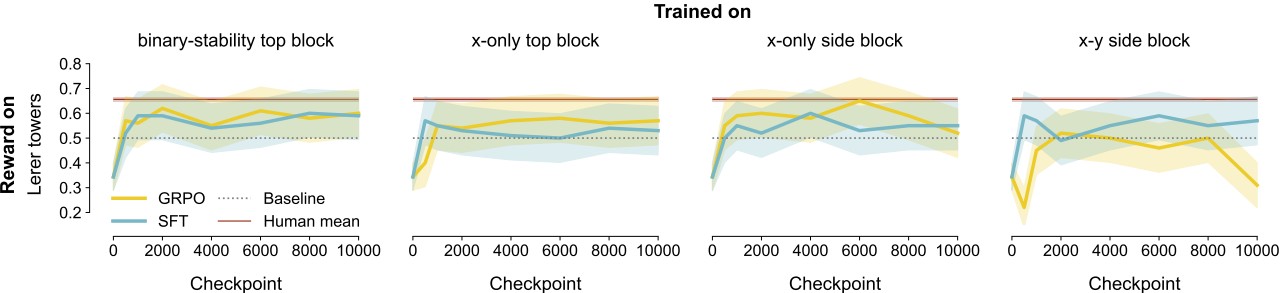

*Figure 3.* Qwen3-VL-8B trained on all four conditions and evaluated on the real images of block towers from Lerer et al. (2016). Crucially, we find some generalization from our synthetic *binary-stability top block* images to real images of block towers. However, we still find that no model fine-tuned on other tasks generalizes to judging the stability of real block towers. Furthermore, we find that even the models post-trained on the binary-stability task do not perform this task at a human level when presented with real images (the red line shows the mean human performance). Error bars show 95% confidence intervals.

## 4.5. Ablations

### 4.5.1. OTHER MODELS

To ensure our results transfer from Qwen3-VL-8B to other models, we repeat all experiments with Qwen2.5-VL-7B. Crucially, for this model, we find that generalization is even more limited (see Fig. 19). This model only transfers from the x-y side block to the x-only side block task — this is to be expected, because x-only is a subset of the x-y task. We perform a number of additional ablations on this model to test if generalization can be improved. We outline these ablations in the following paragraphs.

First, it is possible that the models learn some task-general features, which they can't properly make use of due to task-specific properties. To test this, we take the Qwen2.5-VL-7B model fine-tuned on x-only top block and supervise fine-tune it for a limited number of steps on binary-stability top block (see Section A.9.1 in the Appendix). If the model has learned task-general features over the course of the x-only top block training, it should learn the binary stability task more quickly than the base model — meaning that it should require fewer additional fine-tuning steps to reach good performance. We find that later checkpoints of the x-only trained model very quickly reaches high accuracies in the binary-stability task after just a few steps of SFT, and crucially more quickly than the base model, indicating that the model has learned some transferable features (see Fig. 20 and Section A.9.1 in the Appendix for more details).

Second, the model might need to be trained for longer to unlock generalization. To test whether generalizable physical intuitions could emerge in GRPO models over time, we trained Qwen2.5-VL-7B for up to 48.000 steps. We find that as we exceed 10.000 steps, the model tends to overfit too strongly to the specific reward function of the training task to generalize to other tasks (see Fig. 21 and Section A.9.2 in the Appendix for more details).

Third, as outlined by previous work (Schulze Buschoff et al., 2025a), it is possible that models trained on a single task fail to generalize because they are not exposed to enough variance in their post-training. To check if generalization can be improved by incorporating multiple tasks, we post-train Qwen2.5-VL-7B with GRPO, first on *x-only side block* and then on *binary-stability top block*. Since the model has been trained on the x-only task and also on the top block data set (albeit not at the same time) we would expect this model to generalize well to the *x-only top block* task. We find that this model is still able to perform both tasks it was trained on — however, it does not generalize to *x-only top block* (see Fig. 23 in the Appendix). We also train an SFT model in the same blocked manner, first training it on *x-only side block* and then on *binary-stability top block*. The performance of this model quickly degrades on the task it was trained on first, suggesting that there might be for training with interaction when it comes to learning multiple tasks sequentially. In contrast, a SFT variant with interleaved data, trained on *x-only side block* and *binary-stability top block* at the same time, performs reasonably well on both of its post-training tasks (see Fig. 23 and Section A.9.4 in the Appendix for more details).

Finally, we perform a number additional ablations to ensure our results are not simply artifacts of the training duration, the generation length, or the adapter rank (see Appendix Section A.9). We also evaluate newer and bigger model variants (see Appendix Section A.8), as well as models trained with another RL algorithm in Section 4.5.2 below. Additionally, we also test the post-trained models ability to generalize to real images (see Sections 4.3 and A.10 below).

### 4.5.2. OTHER RL IMPLEMENTATIONS

To ensure that our results transfer to other RL algorithms, we post-train Qwen3-VL-8B and Qwen3-VL-32B with Group Sequence Policy Optimization (GSPO) on the x-only top block task. GSPO replaces the token-level optimization in

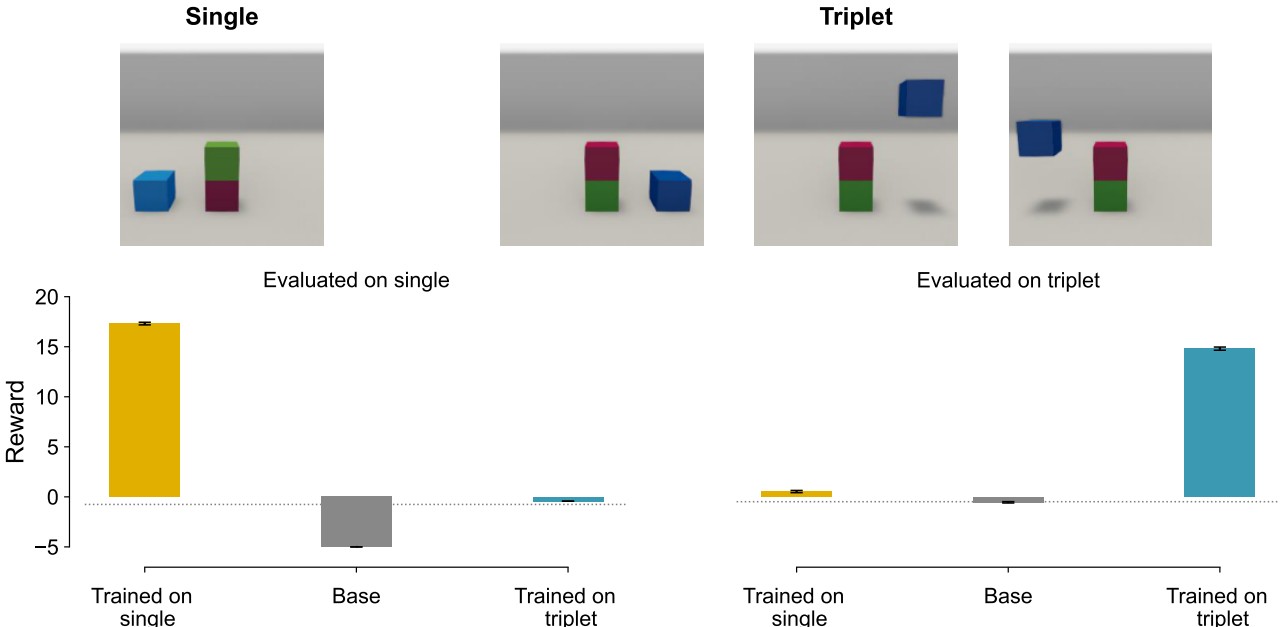

*Figure 4.* Qwen3-VL-8B trained with GRPO on single and multi-step versions of the *x-y side block* task. We find that neither the model trained on the single version, nor the model trained on the triplet version generalize to the other variant of the task. This is despite both tasks sharing the same visual statistics and the same reward function.

GRPO with sequence-level optimization (see Zheng et al. (2025) for more information). We find that the 8B model performs well on their post-training task, and that it shows a similar pattern of generalization to the related binary-stability task as the GRPO model (see Fig. 15). However, we still find that it does not generalize to the other tasks, such as for example x-only side block (see also Section A.8 in the Appendix) — this task is the same as the models' post-training task, only with larger block displacements. If the models learned the mapping between block distance to center and the integer action space, they should in principle also be able to solve this task — but we do not find this, neither in the models trained with interaction nor the models trained with SFT.

The 32B model trained with either GRPO or GSPO does not show meaningful generalization to any condition (see Fig. 16 in the Appendix). In contrast, the 32B model post-trained with SFT generalizes somewhat from the x-only top block post-training task to the binary-stability top block task. This again indicates that there is no reliable benefit of training models with interaction when it comes to learning generalizable physical intuitions.

### 4.5.3. MULTI-STEP TASK

Our formulation of RL is rather limited in that it only includes a single state of the environment, and the agent returns a single action after which the model receives a reward. As such, the model never *sees* the result of actions in the environment — it never receives an updated state of the environment after an action was taken. To remedy this, we add a triplet version of the *x-y side block* condition, where models first see an image of a block displaced next to a block tower. They are then informed of an action that was performed, and they then get a second frame, which is the result of this action. A second action is performed, which results in the final image, upon which the model is asked to output a final action (see Sections A.1 and A.5 in the Appendix for example images and the prompt structure).

We find that Qwen3-VL-8B learns to perform this task well after 10.000 steps of GRPO. However, this model also does not generalize — not even to the single image version of the same task it was trained on. The same is true for the model trained on the single image version: it does not generalize to the triplet version of the same task (see Fig. 4).

## 5. Discussion

Recent evidence has suggested that vision language models (VLMs) do not have robust human-like intuitions about the physical world. For instance, they struggle to reason about the stability of block towers or about cause and effect (Schulze Buschoff et al., 2025a), even when they are fine-tuned on related tasks (Schulze Buschoff et al., 2025b). Humans, on the other hand, have robust intuitions about the physical world, which they learn in part from interacting with their environment.

To capture this aspect of human learning, we trained VLMs on intuitive physics tasks that require building block towers through interaction with an environment. We trained these models using the online reinforcement learning algorithm Group-Relative Policy Optimization (GRPO), and compared it to Supervised Fine-Tuning (SFT), an analogue of offline reinforcement learning (Levine et al., 2020). We then tested these trained models on held-out tower building tasks and on judging the stability of block towers.

Given the relevance of interaction for learning intuitive physics, we defined three hypotheses: (1) that GRPO-trained models would outperform SFT-trained models on held-out instances of the task they were trained on, and (2) that GRPO-trained models would generalize better than SFT-trained models to new tasks, such as judging tower stability.

Our experiments found no evidence in favor of (1). Across all four tasks, both GRPO and SFT post-training led to models performing close to ceiling on held-out test instances. This supports recent results showing that task-specific post-training can make VLMs perform well within the visual intuitive physics problems they are trained on (Balazadeh et al., 2025; Schulze Buschoff et al., 2025b).

For (2), we found that interaction did not confer a clear advantage for generalizing to new tasks. We found that both GRPO and SFT allowed slight generalization within tasks that share the same data distribution (for example from x-only top block to binary-stability top block; note that this is not the case for the older Qwen2.5-VL-7B model).

While we find some traces of generalization, it is very limited and the post-trained models do not transfer to all related tasks. We hypothesized that interaction would be helpful for learning generalizable physical intuitions. However, we do not find clear evidence that training these models with interaction gives them generalizable physical intuitions, nor that it is better than SFT when it comes to generalization to related tasks.

While our decodability analysis showed that the relevant properties for solving the physics tasks, such as binary stability and x-offset, are highly decodable from activations at all intermediate layers in the base, SFT-trained and GRPO-trained models, neither post-training method yielded models that make use this information on out-of-distribution tasks.

**Limitations**   There are several avenues of research that would strengthen the conclusions of this study. We investigate only three model families of sizes 7B, 8B, 11B, 12B and 32B, trained in constrained, single settings. Future work will examine whether our conclusions apply to larger models trained on larger volumes of data and larger varieties of tasks. We also only investigated single-step interactions with

the environment. It remains possible that advantages of interaction only surface when models are able to interact with their environment over long state-action sequences. Future work will investigate practical methods for testing this with modern vision-language models. Finally, we post-trained 4-bit quantized models using PEFT [2], future work should investigate whether our results hold for full fine-tuning.

For future work, our results on multi-task training point toward a path forward: when models are trained with GRPO on two tasks sequentially, they retain the ability to perform both. This is a preliminary but encouraging signal that multi-task training can overcome some of the brittleness we observe in single-task post-training. Future work should explore more diverse training distributions and curriculum-based approaches that gradually introduce more complex physical scenarios. Additionally, the decodability analysis shows that the relevant physical variables are already represented in the models but not recruited for generalization. Future work should investigate auxiliary objectives that encourage models to ground their predictions in the underlying physical variables rather than surface-level shortcuts.

## 6. Conclusion

We hypothesized that through interaction with a simulated environment, vision language models would be able to learn generalizable physical intuitions. However, we find little evidence of this — neither models trained with GRPO nor SFT were able to reliably generalize from their training task to other related tasks. This suggests that these models are not learning true physical intuitions, but rather task-specific shortcuts.

Together, our results suggest that prominent post-training methods are constrained in the ways that they can improve models when it comes to intuitive physics. It remains unclear whether post-training models on specific cognitive tasks is sufficient for developing models that reason about the world in a human-like manner. Developing machine learning models with these abilities may require different pre- and post-training paradigms that go beyond parameter-efficient adaptation.

## Impact Statement

This paper presents work whose goal is to advance the field of machine learning. There are many potential societal consequences of our work, none of which we feel must be specifically highlighted here.

---

[2]Prior work has demonstrated that 4-bit quantization introduces negligible performance degradation compared to full-precision fine-tuning (Dettmers et al., 2023; Frantar et al., 2022; Lin et al., 2024), and that it is the optimal precision for zero-shot accuracy per bit (Dettmers & Zettlemoyer, 2023)

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

# A. Appendix

## A.1. Data examples

### A.1.1. TOP BLOCK DATASET

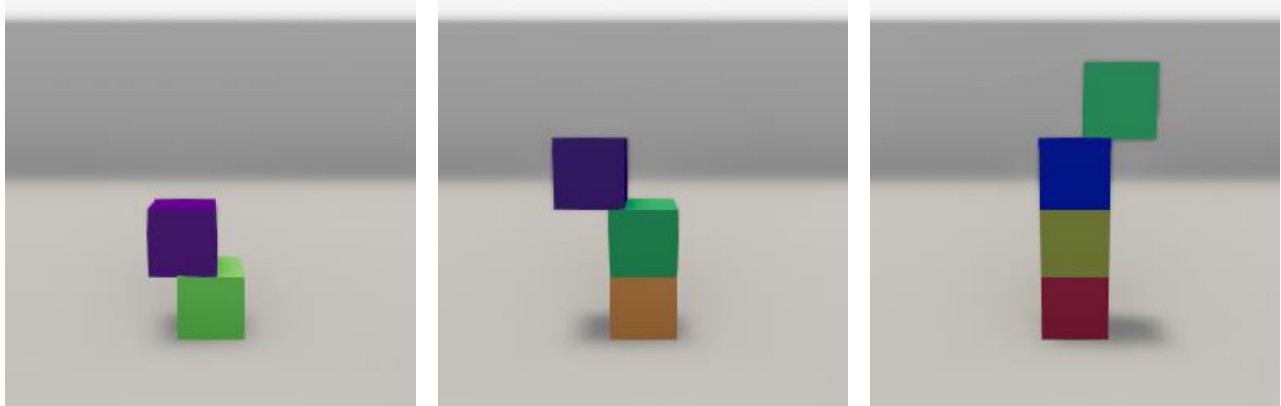

*Figure 5.* Example images for the *top block* dataset. Images feature towers with 2 to 4 blocks with the top block displaced.

### A.1.2. SIDE BLOCK DATASET

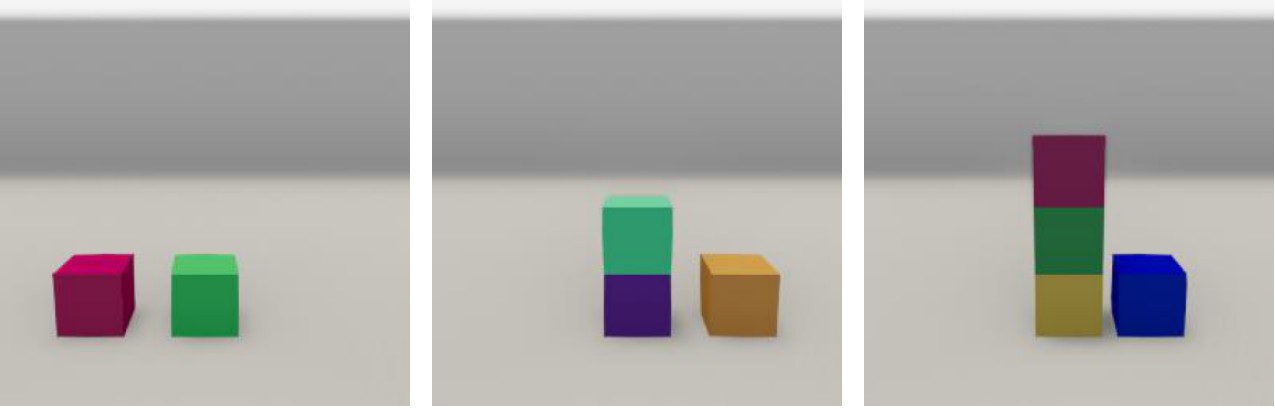

*Figure 6.* Example images for the *side block* dataset. Images feature towers with 1 to 3 blocks with a misplaced block to the side.

### A.1.3. SIDE BLOCK TRIPLET DATASET

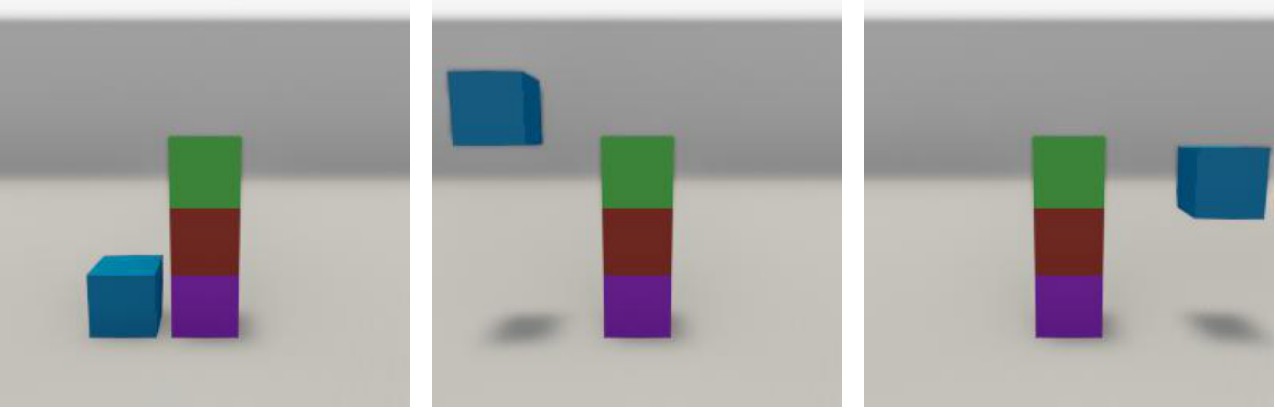

*Figure 7.* Example images for the *side block triplet* dataset. Triplet of a tower with 1 to 3 blocks with a misplaced block to the side.

A.1.4. LERER DATASET

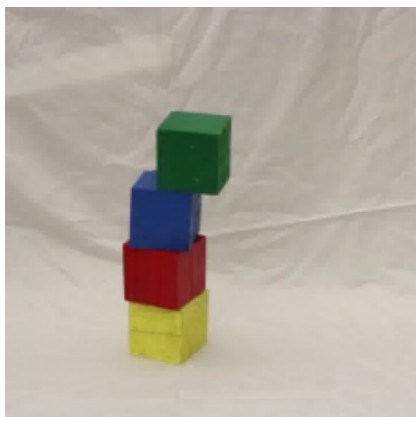 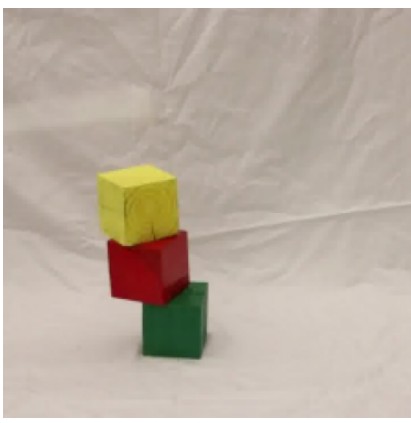 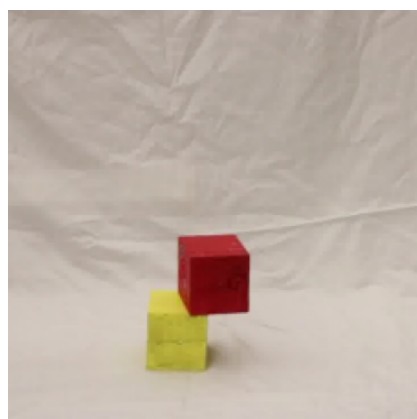

*Figure 8.* Example images for the *Lerer* evaluation dataset. Images are real pictures of block towers with 2 to 4 blocks.

## A.2. Only legal answers

The following table shows the results for all Qwen3-VL-8B models after 10.000 steps of GRPO and SFT on the binary-stability tasks for only legal answer. These tables reflect accuracy for judging the stability of block towers:

| Evaluated on | Trained for 10.000 steps with GRPO on | | | |
| --- | --- | --- | --- | --- |
| | binary-stability top block | x-only top block | x-only side block | x-y side block |
| binary-stability top block | 0.943 | 0.788 | 0.503 | 0.74[3] |
| binary-stability lerer | 0.6 | 0.57 | 0.52 | 0.31 |

| Evaluated on | Trained for 10.000 steps with SFT on | | | |
| --- | --- | --- | --- | --- |
| | binary-stability top block | x-only top block | x-only side block | x-y side block |
| binary-stability top block | 0.969 | 0.624 | 0.596 | 0.506 |
| binary-stability lerer | 0.59 | 0.53 | 0.55 | 0.57 |

## A.3. Reward function visualisation

Below, we visualize the reward functions for the *x-only* and *x-y* tasks. For *x-only*, the reward for answers that result in an unstable tower is calculated as $2 \cdot e^{-d^2} - 2$, where $d$ is the distance on the $x$-dimension. For answers that result in a stable tower, we compute the reward as $20 \cdot e^{-d^2}$ (see Fig. 9 below).

For *x-y*, we compute the euclidean distance between the final position of the moved block and the optimal position on top of the tower. For answers that are above ground but do not result in a stable bigger tower, we calculate the reward as $2 \cdot e^{-d^2} - 2$. For answers that are within the tower, we compute $2 \cdot e^{-d^2} - 4$. And for answers that result in a stable bigger tower, we compute the reward as $20 \cdot e^{-d^2}$ (see Fig. 10 below).

---

[3]5770 out of the 10.000 responses here are illegal

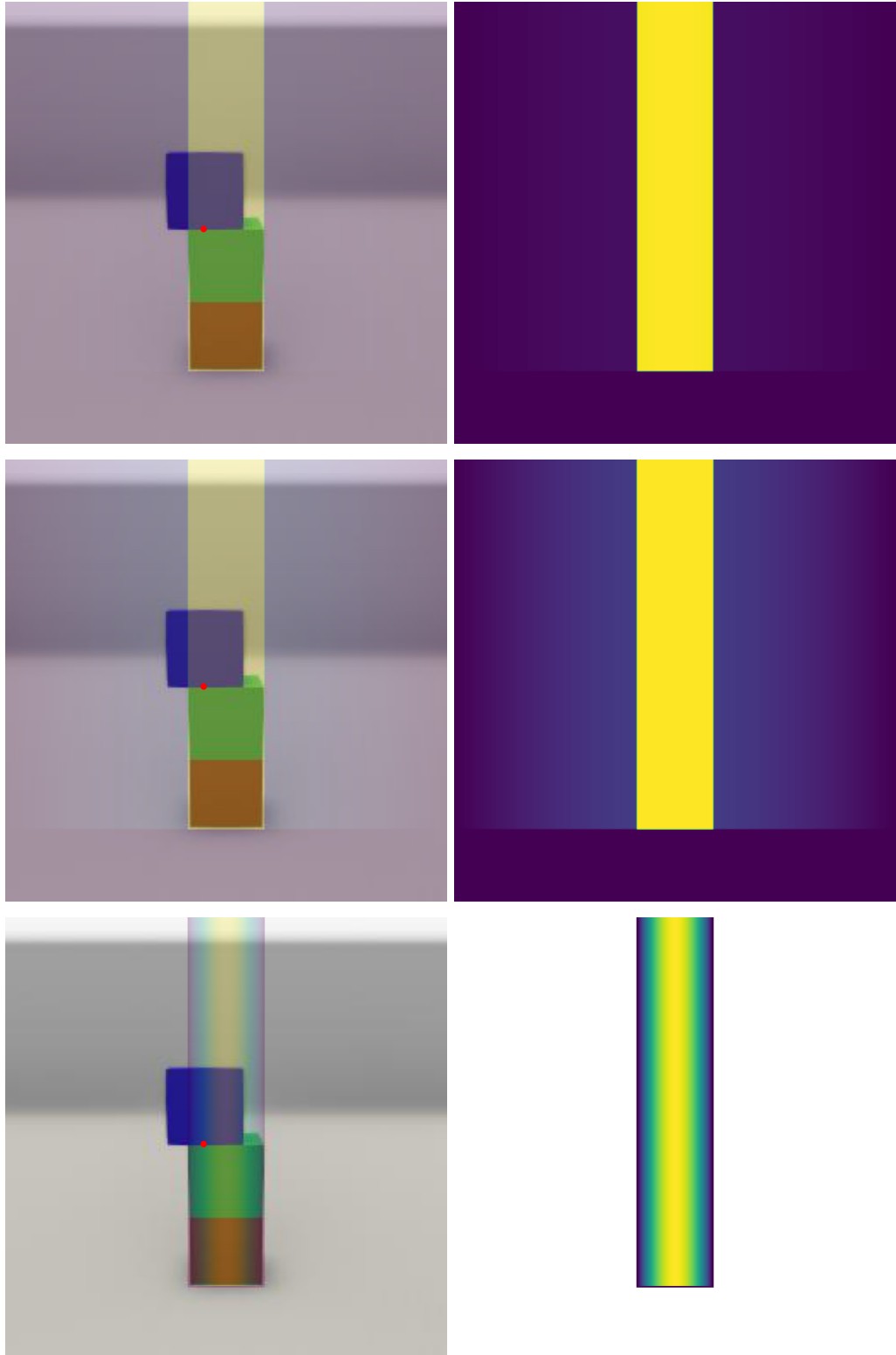

*Figure 9.* Reward function for the x-only task on the top block dataset. The red dot sits at the lower center of the block from which the reward is calculated. The first row shows the non normalized reward values. The second row shows the symmetric-log transformed reward values. The third row shows the log normalized reward values.

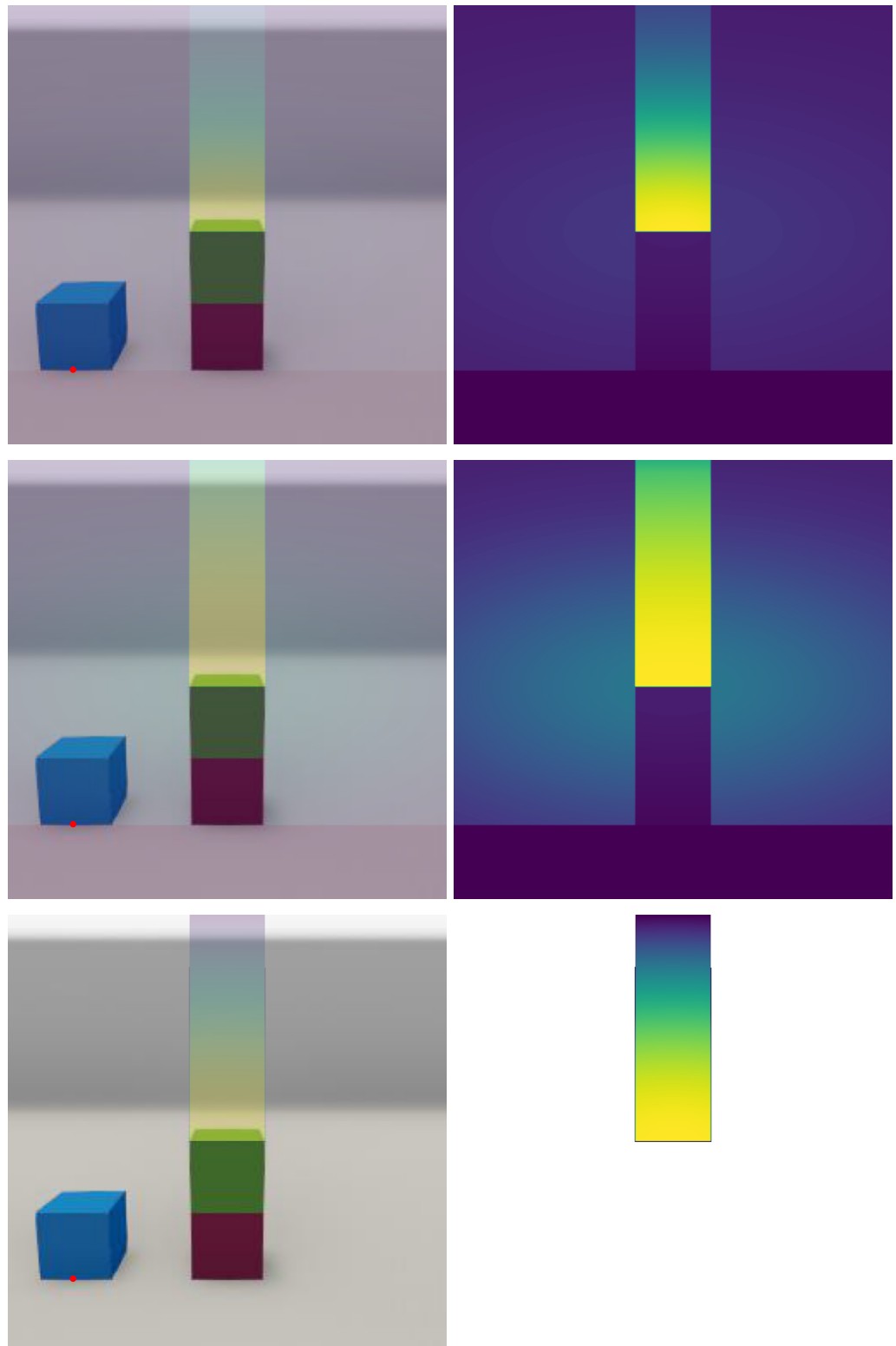

*Figure 10.* Reward function for the x-y task on the side block dataset. The red dot sits at the lower center of the block from which the reward is calculated. The first row shows the non normalized reward values. The second row shows the symlog transformed reward values. The third row shows the log normalized reward values.

## A.4. Training logs

### A.4.1. QWEN3-VL-8B

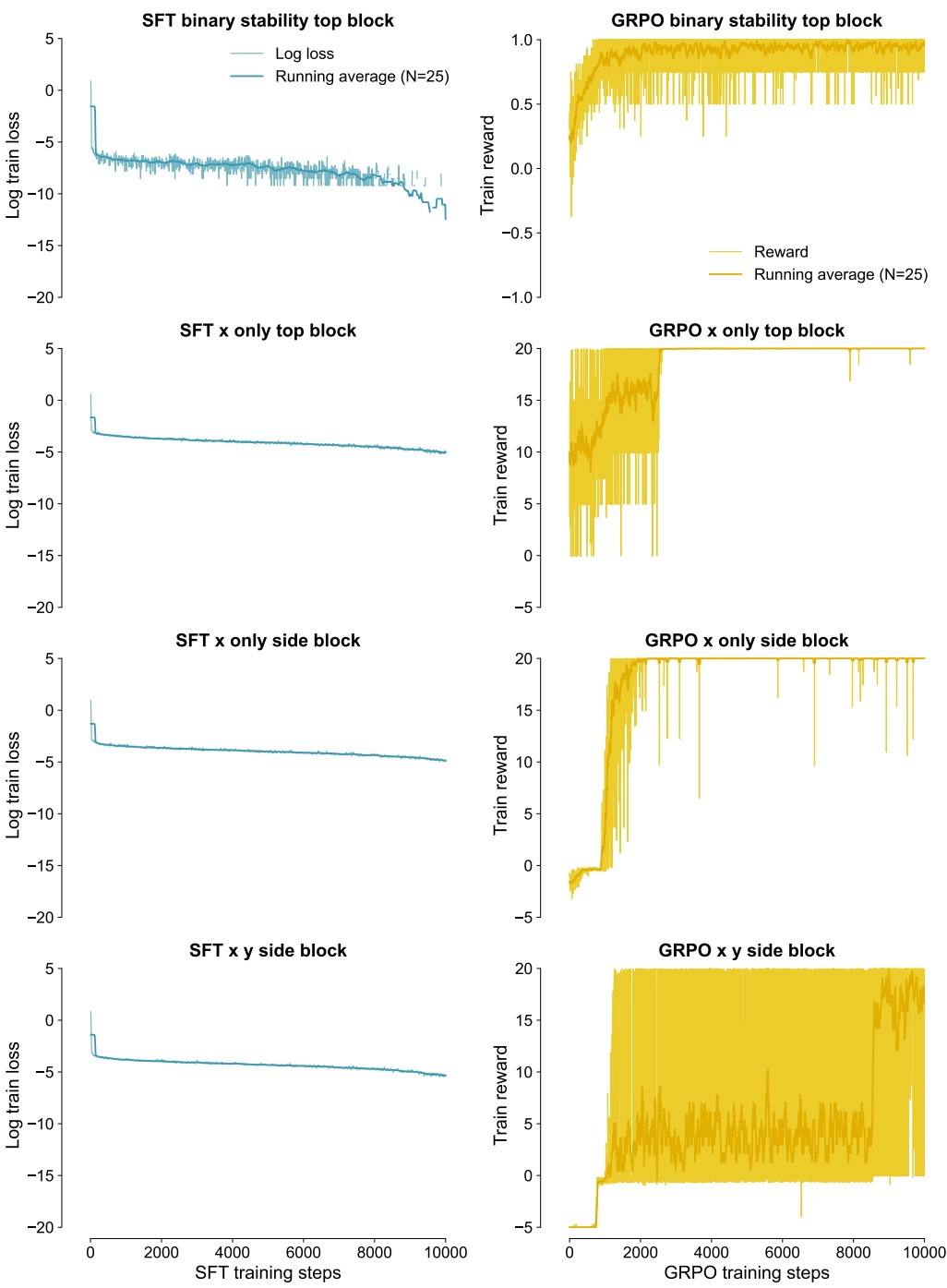

*Figure 11.* Training logs for the SFT (left) and GRPO (right) Qwen3-VL-8B models trained on all datasets. For the SFT models, we show the log loss and a running average with a window of 25. For the GRPO models, we show the mean reward and a running average with a window of 25.

### A.4.2. QWEN2.5-VL-7B

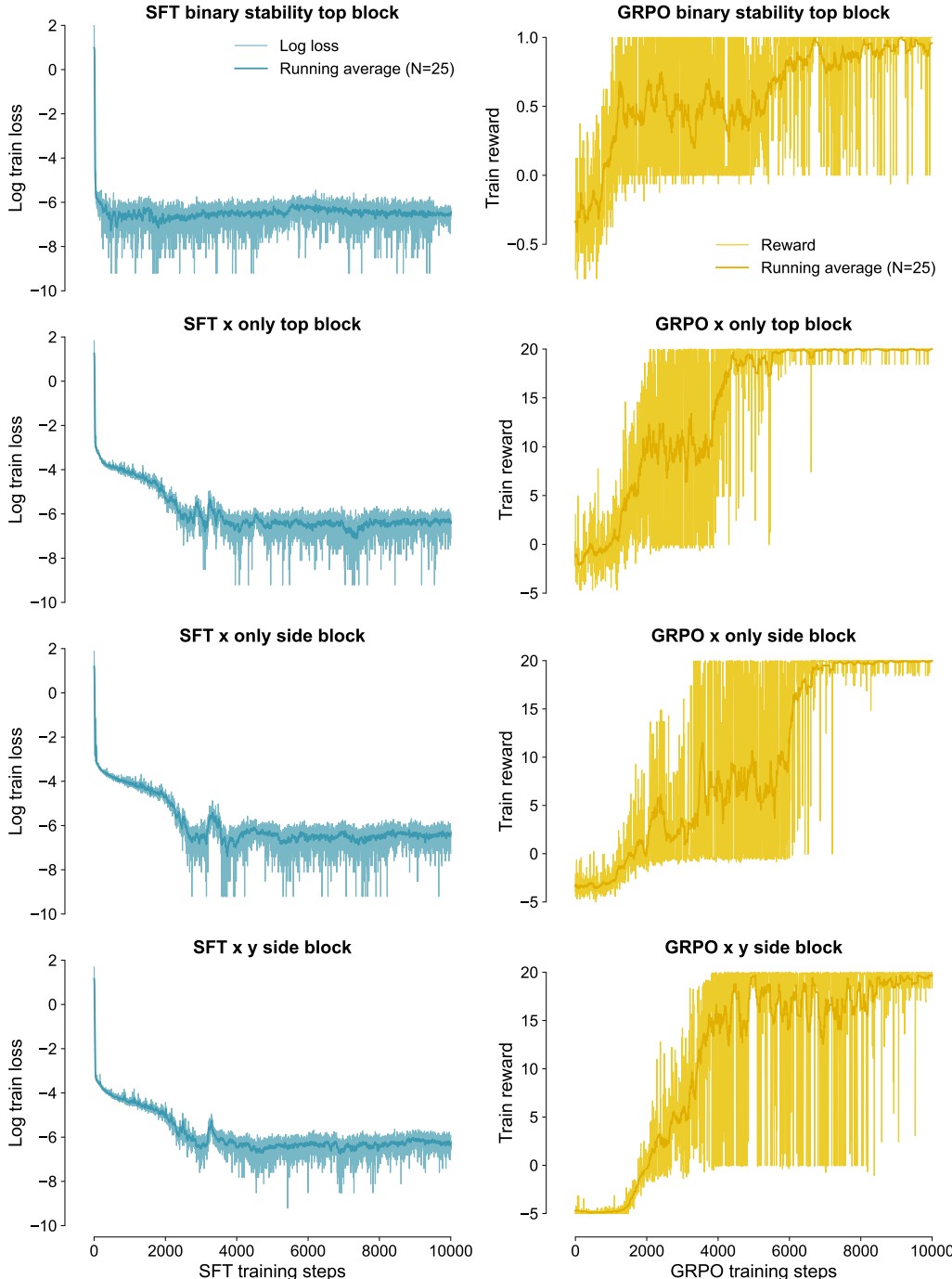

*Figure 12.* Training logs for the SFT (left) and GRPO (right) Qwen2.5-VL-7B models trained on all datasets. For the SFT models, we show the log loss and a running average with a window of 25. For the GRPO models, we show the mean reward and a running average with a window of 25.

## A.5. Task Prompts

We use the following prompt variations for the four tasks depending on action type and dataset (see Fig. 1 for an overview):

For *binary judgment* on the *top block* dataset:

> In the image you see a block tower with a single misplaced block on top. Your task is to determine if this is a stable tower. Respond with Yes if the tower is stable or No if it is not stable. Return your final answer between <answer> </answer>.

For *x-only* on the *top block* dataset:

> In the image you see a block tower with a single misplaced block on top. Your task is to build a stable tower by moving the top block to the most stable position. You can move the top block to the left or right, by responding with an integer between -600 and 600. Return your final answer between <answer> </answer>.

For *x-only* on the *side block* dataset:

> In the image you see a block tower in the centre with a single misplaced block to the side. Your task is to build a stable tower by moving the misplaced block to the most stable position on the top of the tower. You can move the top block to the left or right, by responding with an integer between -600 and 600. Return your final answer between <answer> </answer>.

For *x-y* on the *side block* dataset:

> In the image you see a block tower in the centre with a single misplaced block to the side. Your task is to build a stable tower by moving the misplaced block to the most stable position on the top of the tower. You can move the misplaced block to the left or right and up, by responding with two integers. The first integer should be between -600 and +600 and moves the block left or right. The second integer should be between 0 and +1000 and moves the block up. Return your final answer between <answer> </answer>.

For *binary-stability* on the *Lerer* dataset:

> In the image you see a block tower. Your task is to determine if this is a stable tower. Respond with Yes if the tower is stable or No if it is not stable. Return your final answer between <answer> </answer>.

For *x-only* on the *top block* dataset with reasoning (long generation):

> In the image you see a block tower with a single misplaced block on top. Your task is to build a stable tower by moving the top block to the most stable position. You can move the top block to the left or right, by responding with an integer between -600 and 600. Provide your reasoning between <think> and < /think>. You can think about the problem for as long as you'd like. While thinking, you should robustly verify your solution. Return your final answer between <answer> </answer>.

For *x-y* on the *side block* dataset as triplets:

> user: $< image >$
> In the image you see a block tower in the centre with a single misplaced block to the side. Your task is to build a stable tower by moving the misplaced block to the most stable position on the top of the tower. You can move the misplaced block to the left or right and down or up, by responding with two integers. The first integer should be between -1200 and +1200 and moves the block left or right. The second integer should be between -1000 and +1000 and moves the block down or up. Return your final answer between <answer> </answer>.
> assistant: <answer> 67, 763 </answer>
> user: $< image >$
> In the image you see a block tower in the centre with a single misplaced block to the side. Your task is to build a stable tower by moving the misplaced block to the most stable position on the top of the tower. You can move the misplaced block to the left or right and down or up, by responding with two integers. The first integer should be between -1200 and +1200 and moves the block left or right. The second integer should be between -1000 and +1000 and moves the block down or up. Return your final answer between <answer> </answer>.
> assistant: <answer> -1110, -436 </answer>
> user: $< image >$
> In the image you see a block tower in the centre with a single misplaced block to the side. Your task is to build a stable tower by moving the misplaced block to the most stable position on the top of the tower. You can move the misplaced block to the left or right and down or up, by responding with two integers. The first integer should be between -1200 and +1200 and moves the block left or right. The second integer should be between -1000 and +1000 and moves the block down or up. Return your final answer between <answer> </answer>.

## A.6. Decoding analysis

For the top block dataset, we train linear probes on the representation of the model at each layer to predict the binary stability of a tower and the x-offset of the top block from those representations. Since the image tokens appear before the text tokens, the linear probes only have access to the visual information. We run this process with 10-fold cross validation using 600 images in total. For the binary stability analysis, we train L2 regularized logistic regression models on the representations. For the x-offset analysis, we train linear regression models with spherical Gaussian priors.

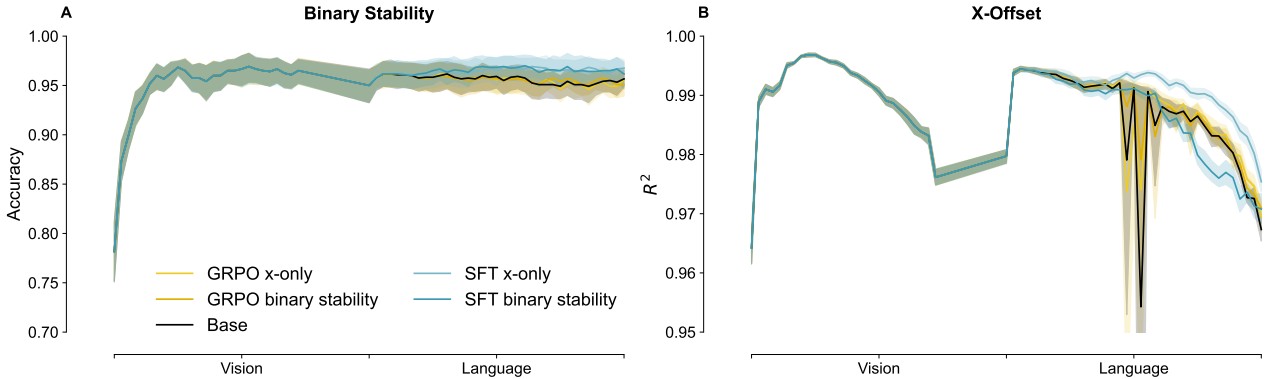

*Figure 13.* Physical property decodability analysis. (A) shows the decodability of the ground truth binary stability for the base model, as well as models post-trained with GRPO and SFT on x-only top block and binary stability top block. (B) shows the decodability of the x-offset of the uppermost block in the top block dataset for the same set of models.

## A.7. Attention maps

To better understand how finetuned models learn to solve our tasks, we compared the attention maps of the fine-tuned models and the base model. More specifically, our goal was to provide a qualitative comparison of how much the last token in the question prompt attends to the different image tokens, throughout the layers of the language model. In Fig. 14, we

show the attention maps averaged across heads for each layer for both the base model and a GRPO model post-trained on x-only top block. As seen in the example below, there is no clear change as a result of the post-training method that would give us a better understanding of the strategy used by the post-trained models.

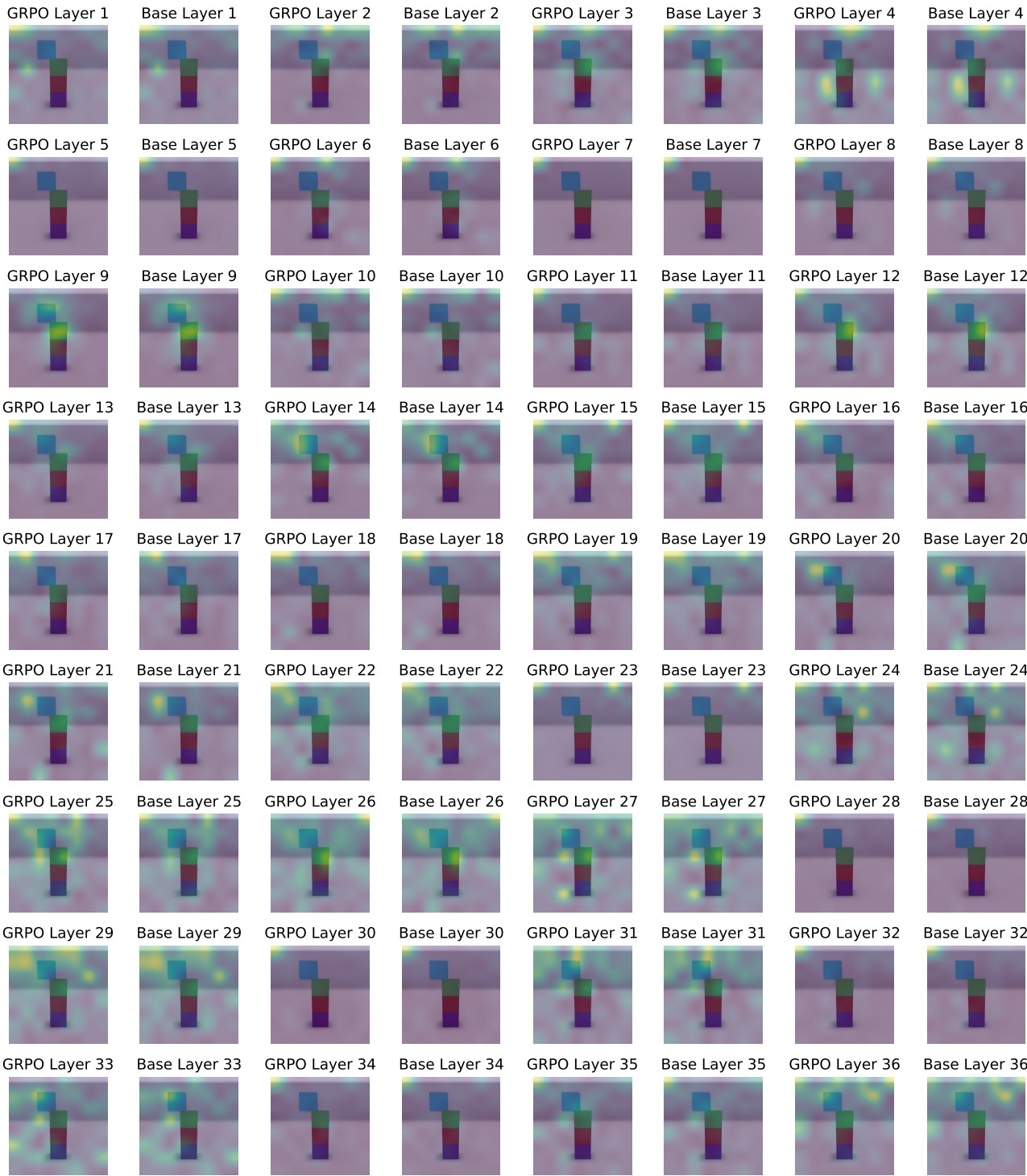

*Figure 14.* Attention maps for the base model and the model post-trained with GRPO on x-only top block. The model is asked if a given tower is stable or not. Attention maps over the different heads are averaged in each layer.

## A.8. Bigger model and another RL implementation

To test whether our results generalize to other models, we train Qwen3-VL-32B with GRPO and SFT on the *x-only top block* task (see Fig. 16 below). To test whether our results generalize to other RL implementations, we also train Qwen3-VL-8B and Qwen3-VL-32B with GSPO on the *x-only top block* task (see Fig. 15 and Fig. 16 below).

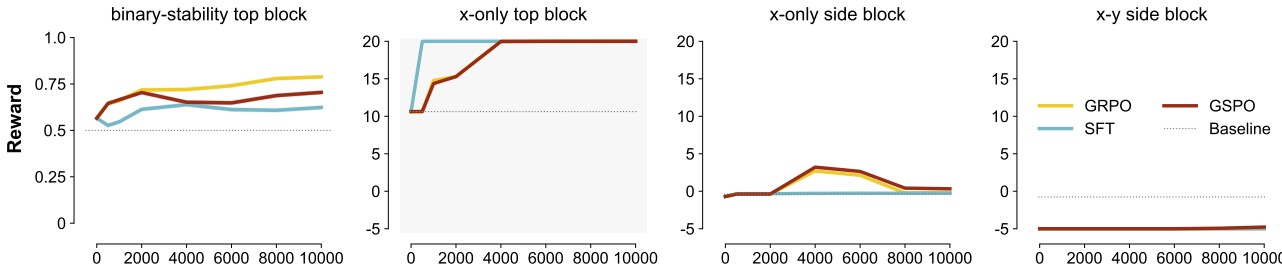

*Figure 15.* Qwen3-VL-8B model trained with GRPO, GSPO, and SFT on the *x-only top block* task. The model also does not generalize from its' fine-tuning task to other related tasks. Noticeably, the model is above chance for the *binary-stability top block* from the get-go and improves slightly over the course of training on the related *x-only top block* task under all post-training regimes.

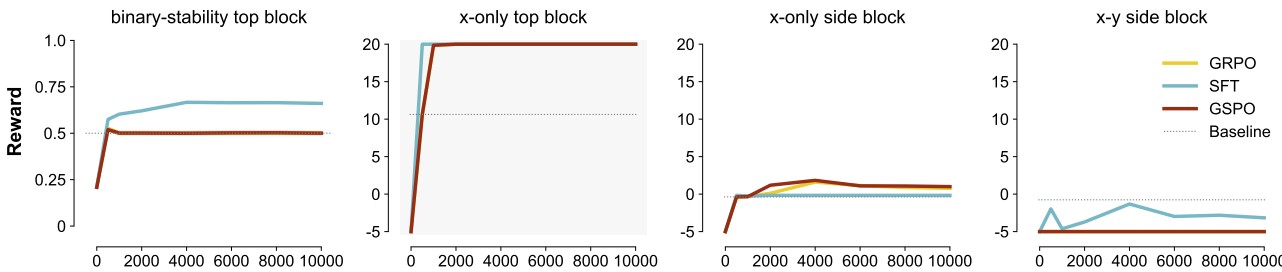

*Figure 16.* Qwen3-VL-32B model trained with GRPO, GSPO, and SFT. The model is trained on the *x-only top block* task. Noticeably, the SFT post-trained model improves slightly over the course of training on the related *binary-stability top block*. In contrast, the GRPO/GSPO post-trained models do not generalize from the *x-only top block* task to the *binary-stability top block* task.

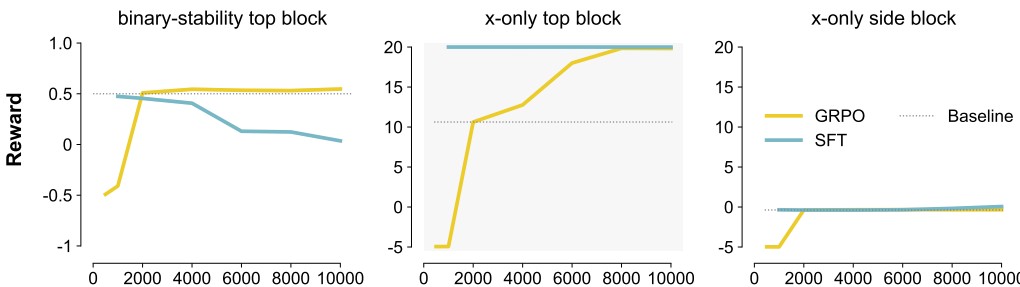

*Figure 17.* Llama-3.2-11B model trained with GRPO on the *x-only top block* task. The model also does not generalize from its' fine-tuning task to other related tasks.

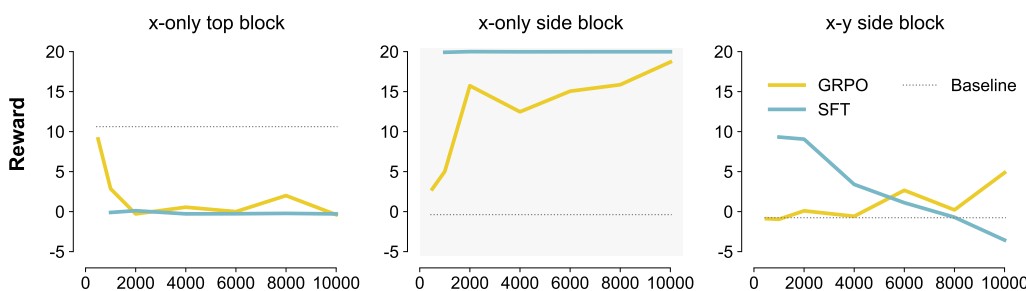

*Figure 18.* Gemma-3-12B model trained with GRPO on the *x-only top block* task. The model also does not generalize from its' fine-tuning task to other related tasks.

## A.9. Additional checks on Qwen2.5-VL-7B

We also fine-tuned Qwen2.5-VL-7B on the full set of task and action combinations (see Fig. 19 for the full set of results). We find that this model, in contrast to Qwen3-VL-8B (see Fig. 2) shows no trace of generalization. We first test whether the model has learned some generalizable understanding after all, that would lead to faster supervised fine-tuning (see Section A.9.1). We then test a number of ablations to test whether other parameter combinations could lead to a better generalizing model (see Section A.9.3).

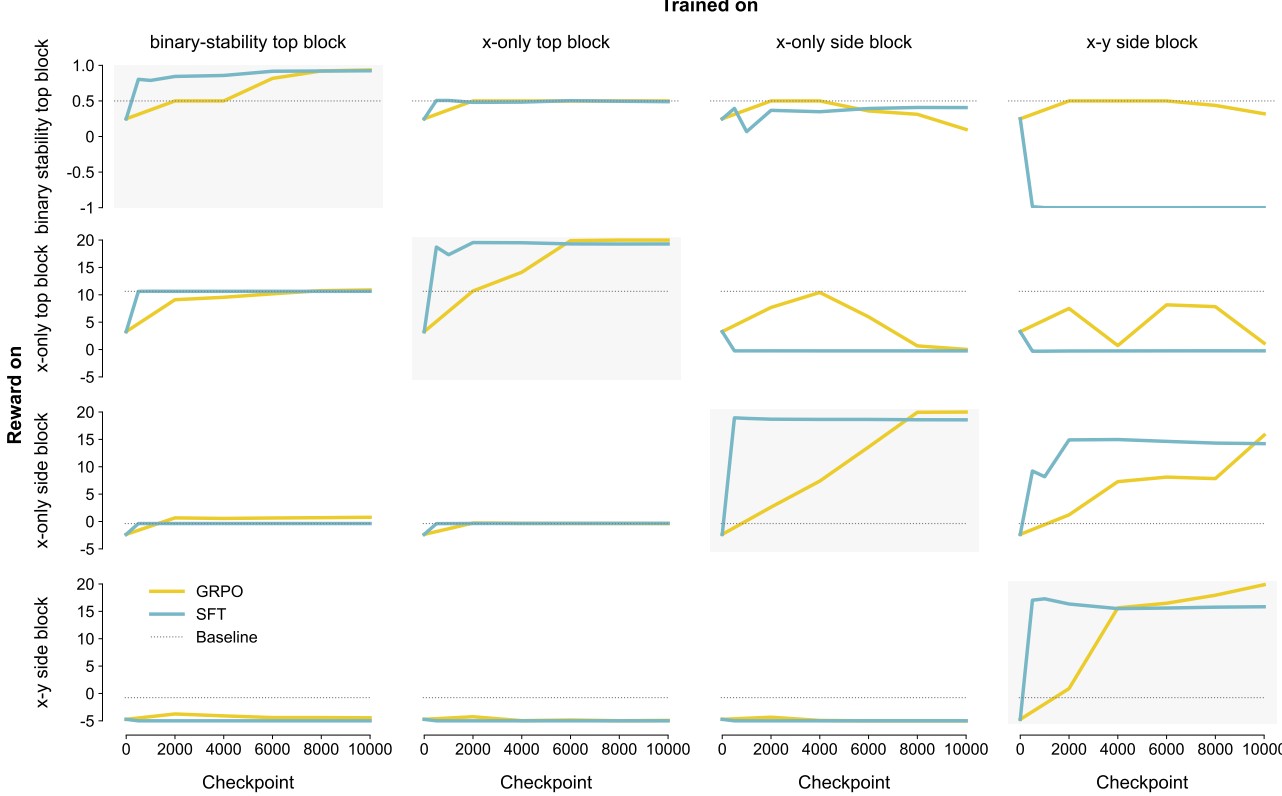

*Figure 19.* Qwen2.5-VL-7B performance by test task and training task. Rows show models evaluated on a given task. Columns show models trained on a given task. The blue and orange lines show the performance of the models trained with SFT and GRPO, respectively. The grey dotted line shows the baseline for the evaluation task. Plots on the diagonal show within-task performance, meaning models are evaluated on the same task they are trained on. All other subplots represent some degree of generalization.

### A.9.1. ADDITIONAL SUPERVISED FINE-TUNING

It is possible that the models have learned some task-general features, but that they fail to perform well on new tasks due to some task-specific properties. To test this, we take checkpoints of the GRPO and SFT x-only top block models, and fine-tune them on the binary stability top block task with some additional steps of SFT. If the models have learned some task-general features, they should learn the binary stability task more quickly than the base model — this means that they should require fewer additional fine-tuning steps to reach good performance.

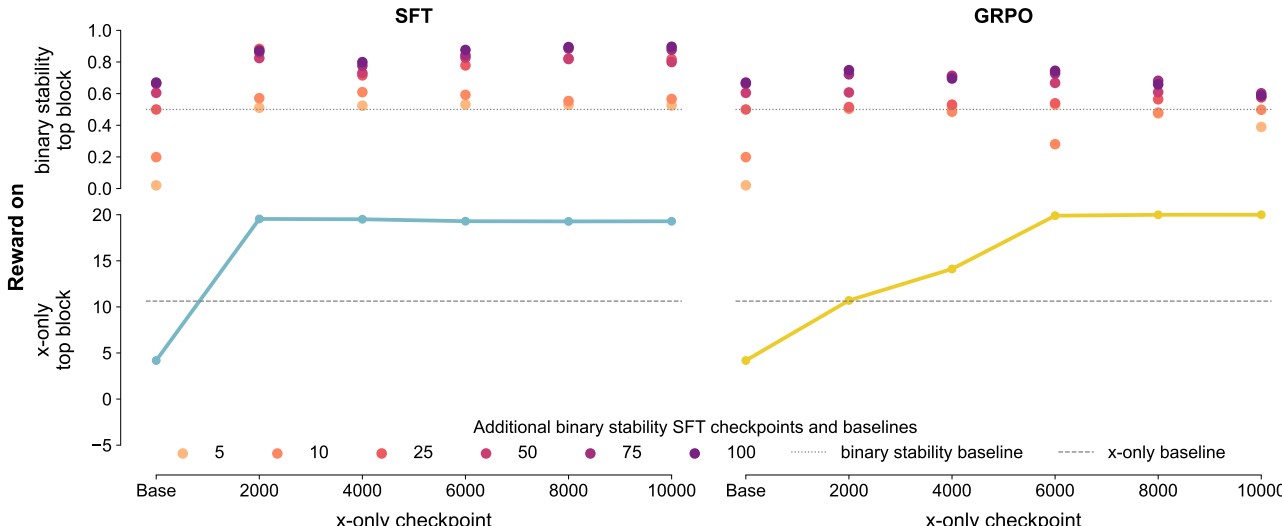

*Figure 20.* Generalization after additional post-training for Qwen2.5-VL-7B. The blue line on the bottom left shows the test performance of the SFT-trained model and the orange line on the bottom right shows that of the GRPO-trained model. Both models were trained on the x-only top block task. For each checkpoint, we take the model trained on this task up to that checkpoint and train it with SFT on the binary stability top block task for up to 100 additional steps. The stacked dots on the top show different checkpoints of the additional binary stability trained models, based on the x-only model at that checkpoint, and the reward they achieve on the binary stability task.

We see that the models very quickly reach high accuracies in the binary stability task after just a few steps of supervised fine-tuning (see Fig. 20). In comparison, the base model fine-tuned with the same number of steps performs less well — while it takes 25 steps for the base model to reach the random baseline for this binary task, all SFT x-only top block checkpoints are already over the baseline after 5 additional SFT steps on binary stability. This is likely in part because the base model has not yet learned to format its answers correctly, whereas all post-trained models have experience with the correct answer format. However, after 100 steps of SFT on binary stability, the base model only returns legal answers but still achieves a lower accuracy than the post-trained models with the same number of additional SFT steps, meaning formatting can not explain the whole performance gap.

We see that SFT post-trained models in general achieve slightly higher accuracies than their GRPO counterparts after 100 additional steps of SFT on the binary stability task: the models trained with SFT on x-only top block up to 2000, 4000, 6000, 8000, and 10000 steps achieve accuracies of 0.778, 0.844, 0.829, 0.806, and 0.800. In contrast, the same checkpoints for the GRPO trained models get accuracies of 0.749, 0.695, 0.745, 0.659, and 0.588. We also see that GRPO models from earlier x-only checkpoints are able to achieve higher accuracies on the binary stability task than later checkpoints.

### A.9.2. LONGER TRAINING HORIZON

We find that training models with GRPO for longer only leads to overfitting to the specific training task (see Fig. 21). As training exceeds 10.000 steps, models tend to overfit too strongly to the specific reward function of the training task to generalize to other tasks — while we still saw some generalization for the x-y side block trained model to the x-only side block task, this disappears as the models are trained for longer. The results reported above all use a restricted generation length due to resource constraints.

To test whether generalizable physical intuitions could emerge in GRPO models over time, we trained Qwen2.5-VL-7B for up to 48.000 steps. We find that as we exceed 10.000 steps, the model tends to overfit too strongly to the specific reward function of the training task to generalize to other tasks — while we still saw some generalization for the *x-y side block* trained model to the *x-only side block* task, this disappears as the models are trained for longer.

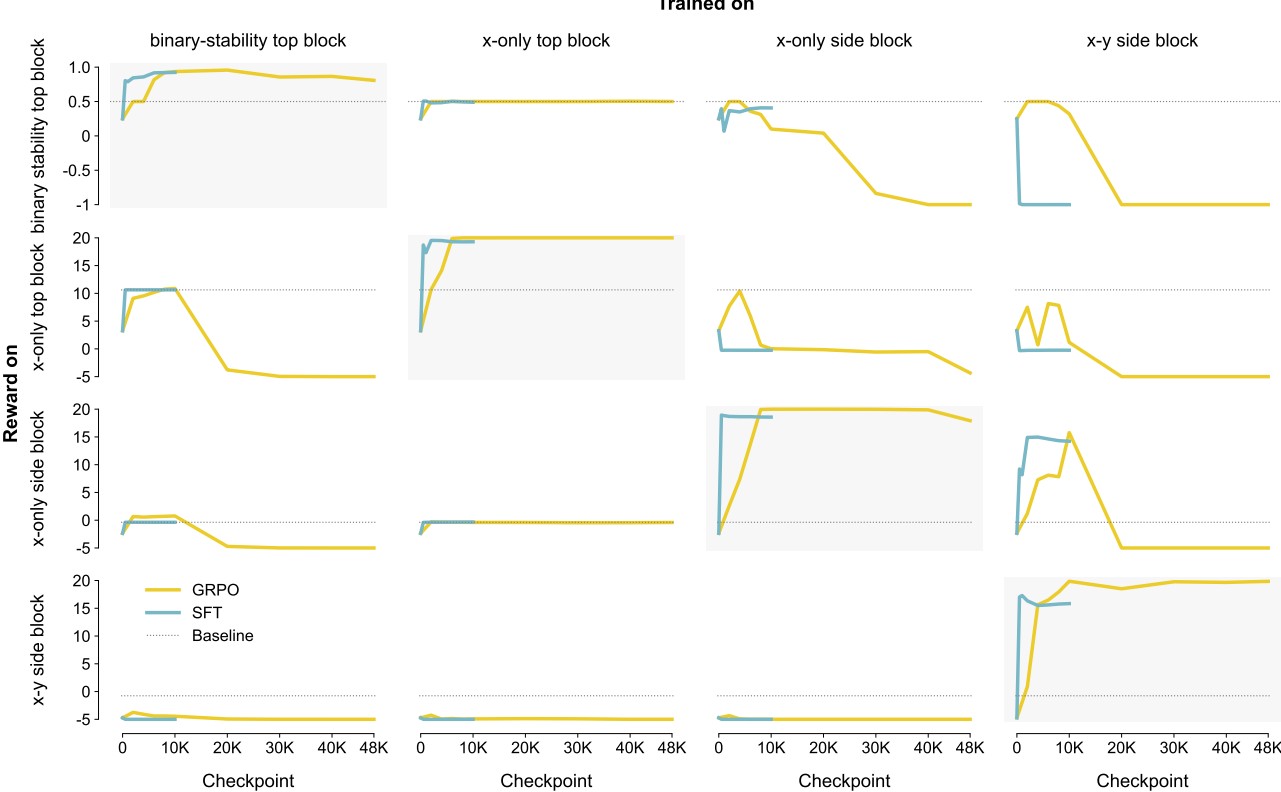

*Figure 21.* Performance for a longer horizon. We show the same plot as 19 but with results for up to 48K training steps for the Qwen2.5-VL-7B GRPO models. Performance is shown for each combination of test task and training task. Rows show models evaluated on a given task. Columns show models trained on a given task. The blue and orange lines show the performance of the models trained with SFT and GRPO, respectively. The grey dotted line shows the baseline for the evaluation task. Plots on the diagonal show within-task performance, meaning models are evaluated on the same task they are trained on. All other subplots represent some degree of generalization.

### A.9.3. TRAINING ABLATIONS

To test if allowing the model to reason about the task for longer improves generalization, we train a model on the *x-only top block* task with a longer generation length (see A.5 for the reasoning prompt). However, as shown in Fig. 22, we find that this model also does not generalize to the other tasks.

The results we report use a default rank of 16 for all models. To make sure that this does not cause the models to overfit to our task specifically, we train models on the *x-only top block* task using ranks of 1 and 8 instead. We find that these models show the same failure to generalize. While models of all ranks learn to perform well on their training task, they do not generalize to any other task (see Fig. 22 in the Appendix).

Additionally, to ensure that the models do not suffer from overfitting the vision encoder, we train a model with the standard rank of 16 but without fine-tuning the vision encoder. We find that this model shows a similar performance over all tasks as the model with vision fine-tuning, again not generalizing to other related tasks from the training task (see Fig. 22 in the Appendix).

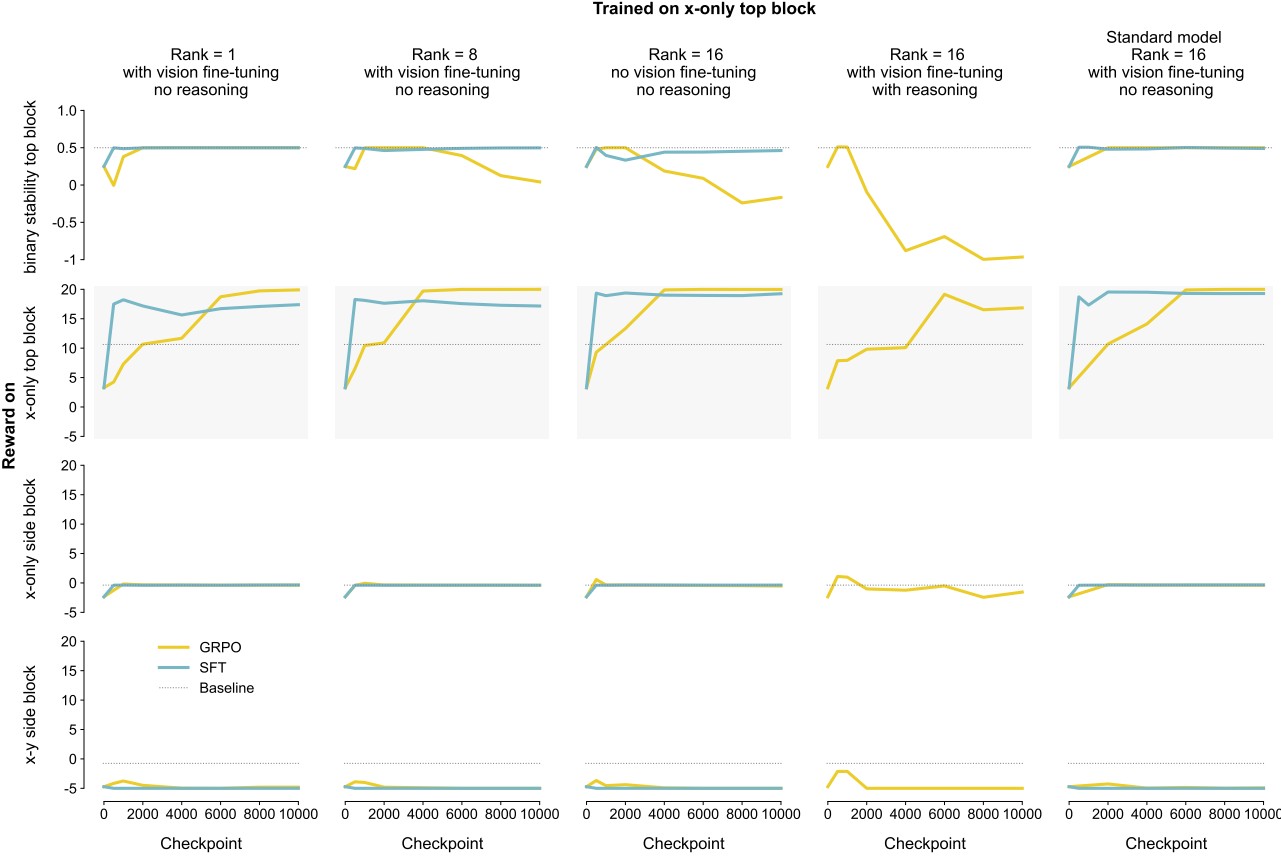

*Figure 22.* Ablation checks. All models are trained on x-only top block but they have lower ranks (1 and 8 compared to 16) or are trained without fine-tuning of the vision encoder or they are trained with reasoning (a larger generation length). All ablations learn to perform well on the task they are trained on (shown in the second row). However, all models fail to generalize to other related tasks — just as the standard model we used throughout our experiments (last column).

### A.9.4. BLOCKED AND INTERLEAVED JOINT TRAINING

To test whether models could generalize if they are exposed to multiple tasks at the same time, we trained models on two tasks: *x-only side block* and *binary-stability top block*. Since this model has seen the *x-only* task and also the *top block* data set (albeit not at the same time), it should be able to generalize to the *x-only top block* data set. We show results for this in the figure below.

We find that the GRPO model that has been trained on both tasks can still perform both tasks. The model has some trouble keeping the correct formatting for the first task block it was trained on, but filtering only legal answers reveals that it still retains the capacity to solve it. The SFT model on the other hand quickly degrades in performance on the task it was trained on first. This indicates some benefit of GRPO when training models on multiple tasks successively. However, the joint SFT model that is trained on both tasks at the same time, in contrast to in a blocked manner, can overcome this shortcoming, performing reasonably well on both of its post-training tasks.

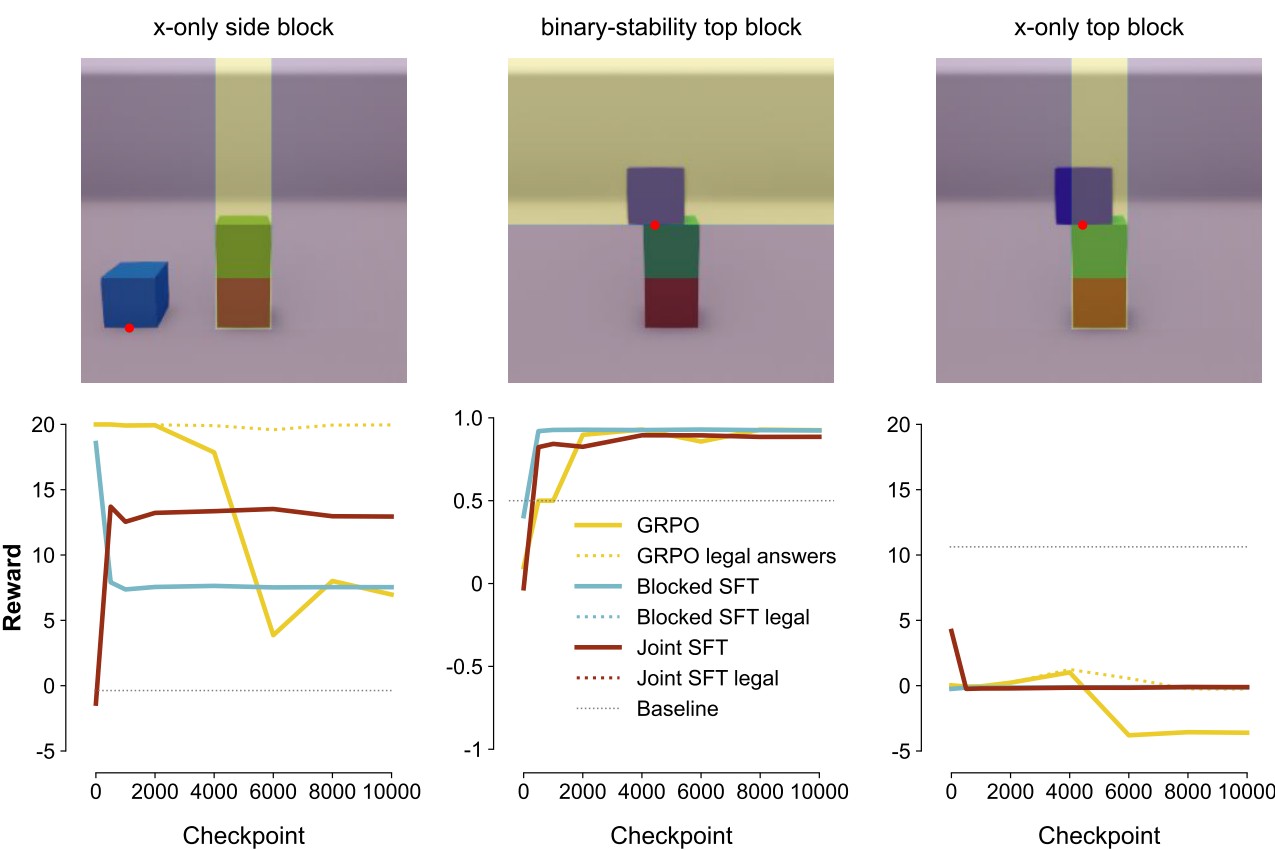

*Figure 23.* Blocked joint training. The model was first trained for 10.000 steps on *x-only side block*. It is then trained for 10.000 steps on *binary-stability top block* and performance is shown here over these 10.000 steps. The model forgets the proper formatting of responses for the initial *x-only side block* task (see continuous line on the left), but legal answers still perfectly solve the task (see dotted line on the left). The model can perform both tasks it was trained on, however it does not generalize to the *x-only top block* task, even though it has seen both the *x-only* task and also the *top block* data set (albeit not at the same time).

## A.10. Generalization to real images

To test whether models could generalize to the same task but presented in natural images, we first test whether any model (Qwen3-VL-8B, Qwen2.5-VL-7B, Qwen3-VL-32B) post-trained with any method (GRPO, SFT, GSPO) on the x-only top task can transfer to predicting binary stability for real images of block towers from Lerer et al. (2016). Both tasks require models to infer the offset of the top block. However, we find that no model generalizes to this task.

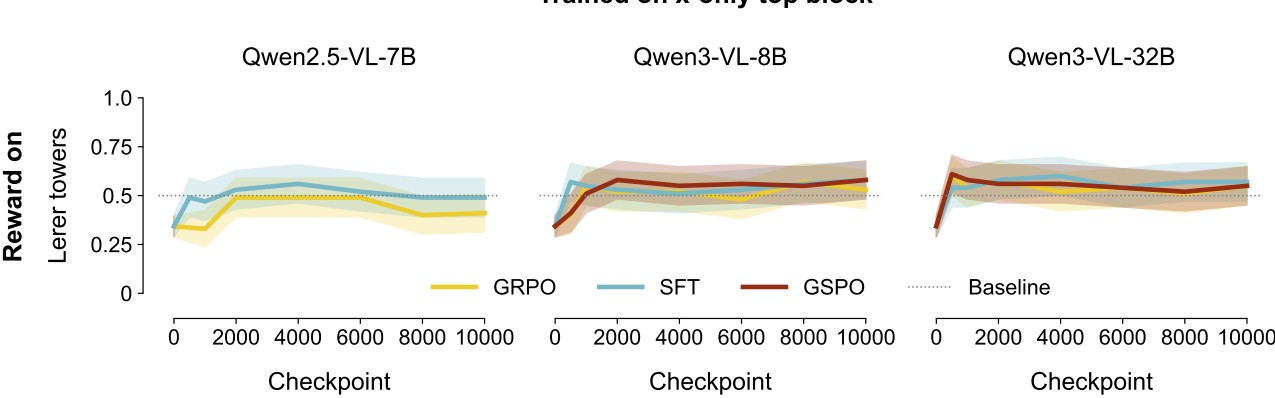

*Figure 24.* All models trained with GRPO, GSPO, and SFT on the *x-only top block* task, evaluated on the real block towers from Lerer et al. (2016). While we still found some generalization from post-training on our *x-only top block* task to our *binary-stability top block* task, the models do not generalize to this external task. Error bars show 95% confidence intervals.

We also test whether any model Qwen2.5-VL-7B model post-trained with either GRPO or SFT can transfer to predicting binary stability for real images of block towers from Lerer et al. (2016). Here, the binary-stability top block model is trained on exactly the same task, only with artificial block towers instead of real images of block towers. We again find that no model generalizes to this task, even the model trained on the similar binary-stability task

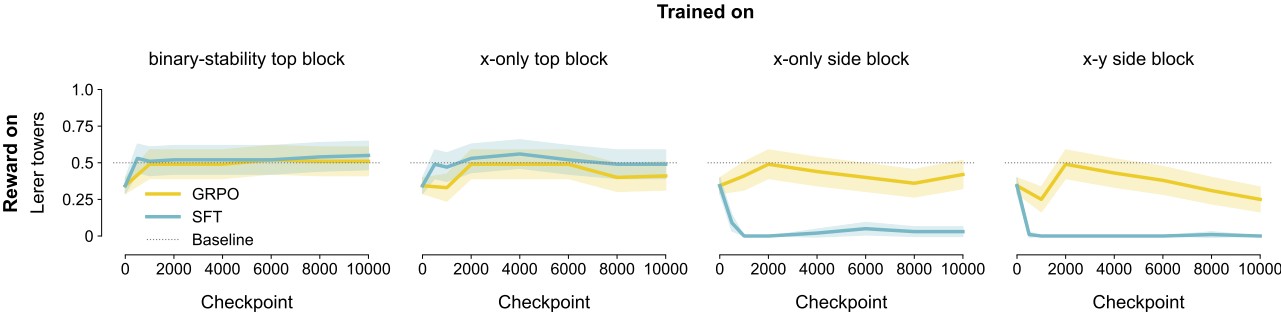

*Figure 25.* Qwen2.5-VL-7B models trained on all four tasks, evaluated on the real block towers from Lerer et al. (2016). We again find that no model performs well on this task — even the model trained on our similar *binary-stability top block* task. Error bars show 95% confidence intervals.

## A.11. Prompt sensitivity analysis

To make sure that our results are not artifact due to the nature of the specific prompts, we perform a prompting analysis. We compare the performance of the *x-only* and **binary-stability** fine-tuned models on *top block* given the prompt they are trained on (variation 0), with 4 other variations that are listed below. We test the models on the same 1000 images from the top block dataset for each of the prompt variations:

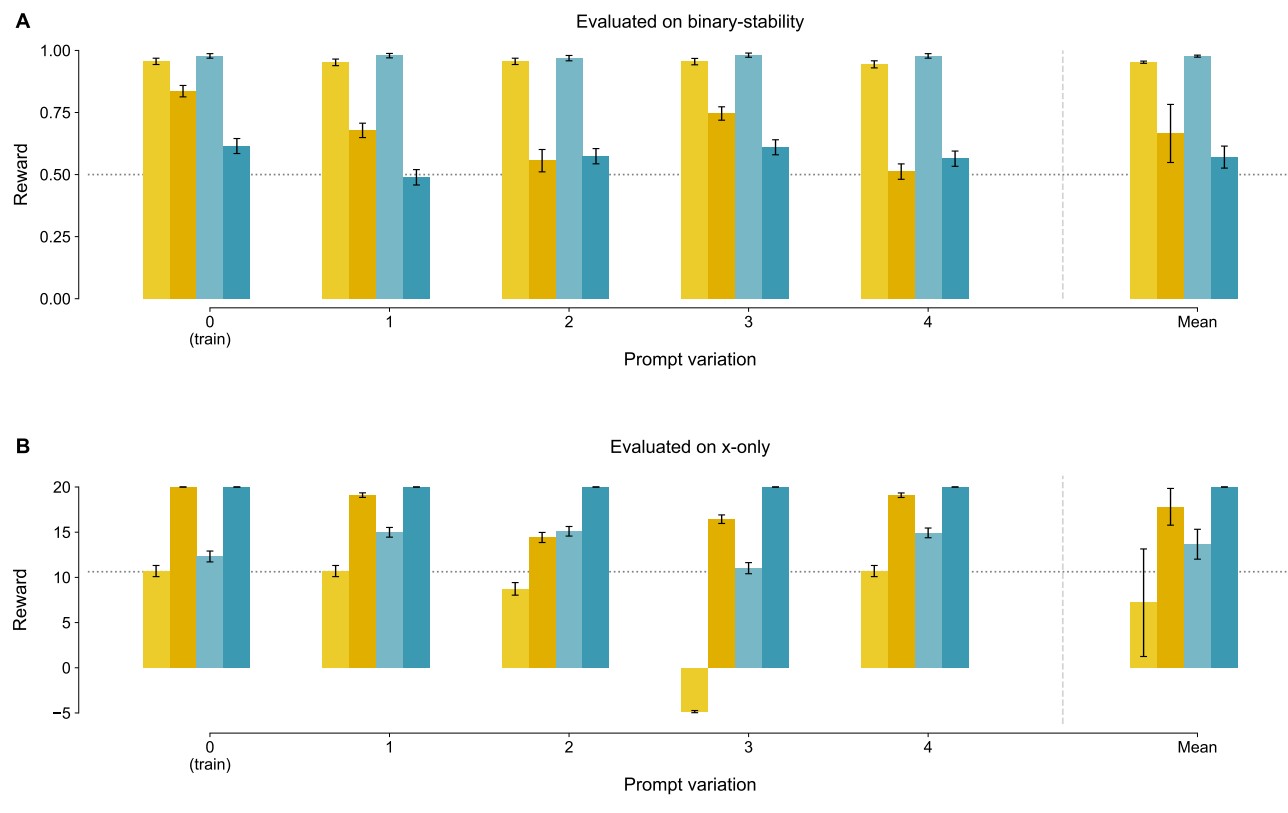

*Figure 26.* Prompt sensitivity analysis for Qwen3-VL-8B models trained on binary-stability top block, x-only top block, and the base model. We find that the fine-tuned models are largely robust against prompt variations. Performance of the generalizing models changes slightly depending on the evaluation prompt.

BINARY-STABILITY PROMPTS

**0** (Train): In the image you see a block tower with a single misplaced block on top. Your task is to determine if this is a stable tower. Respond with *Yes* if the tower is stable or *No* if it is not stable. Return your final answer between `<answer>` `</answer>`.

**1**: The image shows a block tower with a misplaced block on top. Is this tower stable? Answer *Yes* or *No*. Return your final answer between `<answer>` `</answer>`.

**2**: Look at the block tower in the image. The topmost block is not in its default position. Will this tower remain standing? Respond with *Yes* if stable or *No* if unstable. Return your final answer between `<answer>` `</answer>`.

**3**: You are shown a block tower with a displaced top block. Predict whether this configuration is stable (the tower will not fall) or unstable (the tower will fall). Respond with *Yes* for stable and *No* for unstable. Return your final answer between `<answer>` `</answer>`.

**4**: A block tower is shown with its top block misplaced. Will the tower stand or fall? Answer *Yes* if it stands or *No* if it falls. Return your final answer between `<answer>` `</answer>`.

X-ONLY PROMPTS

**0** (Train): In the image you see a block tower with a single misplaced block on top. Your task is to build a stable tower by moving the top block to the most stable position. You can move the top block to the left or right, by responding with an integer between $-600$ and $600$. Return your final answer between `<answer>` `</answer>`.

**1**: The image shows a block tower whose topmost block is misplaced. Stabilize the tower by moving the top block left or right. Respond with a single integer in the range $[-600, 600]$ indicating the required displacement. Return your final answer between `<answer>` `</answer>`.

**2**: Look at the block tower in the image. The top block is out of position. How far should it move left or right to make the tower stable? Give your answer as a whole number between $-600$ and $600$. Return your final answer between `<answer>` `</answer>`.

**3**: You are presented with an image of a block tower in which the uppermost block is displaced. Your objective is to determine the horizontal displacement required to position the top block optimally for tower stability. Express your answer as an integer in the interval $[-600, 600]$, where negative values indicate leftward movement and positive values indicate rightward movement. Return your final answer between `<answer>` `</answer>`.

**4**: A block tower has its top block misplaced. Move it left or right to stabilize the tower. Respond with an integer between $-600$ and $600$. Return your final answer between `<answer>` `</answer>`.

