# OpenReview forum: "Can Vision Language Models Learn Intuitive Physics from Interaction?"
_ICML.cc/2026/Conference — ICML 2026 regular_

### Official Review · Reviewer_SyFm · 2026-02-25

**Soundness:** 2
**Presentation:** 2
**Significance:** 4
**Originality:** 3
**Overall Recommendation:** 5
**Confidence:** 3

**Summary:**

This paper discussed that whether VLM can have human-like, generalizable intuitions of physics, but finds limited evidence under the tested settings.

**Compliance With Llm Reviewing Policy:**

Affirmed.

**Final Justification:**

I remain positive about this submission. The question the paper studies whether VLMs can learn intuitive physics is meaningful. Although the experimental design is not perfect, it is still sufficient to support the paper’s main conclusions. The authors report a negative result, and the breadth and detail of the experiments make this result practically meaningful. In addition, my initial concerns about soundness, mainly regarding the unclear dataset/experiment scale and several experimental details, were addressed in the rebuttal. This has increased my confidence in the paper’s soundness.
Therefore, I have raised my score to a 5.

**Key Questions For Authors:**

1. What is the scale of data in the experiment?
1. The 'interaction' is a regression problem, but the model is evaluated through an additional mapping to text-token outputs (which the author also mentioned). So, could the authors design experiments to disentangle these factors to test whether these contributes to the bottleneck?
1. Could the negative results be partly due to limitations of the Qwen model family? In some tasks, Qwen appears to exhibit different behavior[1]

[1] Shao, R., Li, S. S., Xin, R., Geng, S., Wang, Y., Oh, S., ... & Zettlemoyer, L. (2025). Spurious rewards: Rethinking training signals in rlvr. arXiv preprint arXiv:2506.10947.

If more sound and depth insights are made, I'll raise the score as I believe the topic is significant.

**Limitations:**

yes.

**Strengths And Weaknesses:**

- Soundness: The paper claims that SFT and RL are not sufficient for enabling VLMs to acquire physical intuition. However, the scale and methodology of the experiments are closer to a demo, which makes the results quite limited. Larger-scale experiments would improve the soundness of the paper.

- Presentation: The paper is well structured. A small suggestion: its main focus is interaction, while the judge task is not designed to involve interaction. It may be better to use more appropriate wording to describe this aspect.

- Significance: This paper raises a question that has received strong attention from the community. The problem is important and has meaningful potential impact.

- Originality: The experimental design is interesting and novel.

---

> ### Author Rebuttal · Authors · 2026-03-31
>
> Dear reviewer SyFm,
>
> Thank you for your review. We appreciate your assessment of us addressing a problem that is "important and has meaningful potential impact" and that our "experimental design is interesting and novel". We will address the concerns you raised below:
> \
> \
> **The scale and methodology of the experiments are closer to a demo, which makes the results quite limited. Larger-scale experiments would improve the soundness of the paper.**
>
> We thank the reviewer for this comment and hope to convince them that the scope of our study goes well beyond a demo. Training and evaluating these models with RL is computationally expensive — each model is trained on 10,000 samples per task and evaluated across all task and dataset combinations at every checkpoint. We report results for three model sizes (Qwen2.5-VL-7B, Qwen3-VL-8B, Qwen3-VL-32B) and three training algorithms (GRPO, SFT, GSPO), with ablations covering LoRA adapter rank, vision encoder fine-tuning, reasoning, training horizons up to 48,000 steps, and blocked vs. interleaved multi-task training. In response to reviewer feedback, we have further added a multi-step variant of the x-y side block task and results for additional model families (Gemma3-12B and Llama3.2-11B). The consistency of our findings across all these configurations gives us confidence in the conclusions we draw.
> \
> \
> **The paper is well structured. A small suggestion: its main focus is interaction, while the judge task is not designed to involve interaction. It may be better to use more appropriate wording to describe this aspect.**
>
> We thank the reviewer for this suggestion. We interpret this as noting that stability judgment is non-interactive by nature, and therefore an unusual evaluation for an interaction-based paradigm. This is intentional: stability judgment is one of several transfer tasks we use, and we test every model on every other task in our suite, spanning different visual stimuli, action spaces, and underlying physical variables (Figures 1 and 2). The answer, as our results consistently show, is that models do not transfer across any of these combinations, suggesting they learn brittle, task-specific shortcuts rather than generalizable physical intuitions. If the reviewer had a different concern in mind, we would be happy to address it.
> \
> \
> **What is the scale of data in the experiment?**
>
> We train the models on 10,000 images for each dataset and task combination. The GRPO models generate 16 responses per sample, over which the reward is calculated, and models are evaluated on a separate set of 10,000 images per task. We have added this information explicitly to the methods section.
> \
> \
> **The 'interaction' is a regression problem, but the model is evaluated through an additional mapping to text-token outputs (which the author also mentioned). So, could the authors design experiments to disentangle these factors to test whether these contributes to the bottleneck?**
>
> Our existing analyses already go a substantial way toward disentangling physical understanding from output formatting. Our decodability analysis (Section 4.4, Fig. 11) shows that the physical quantities required to solve each task are highly decodable from model activations across all post-training conditions — including those where behavioral transfer fails. The relevant information is present in the model's representations; it is simply not recruited for out-of-distribution outputs. The additional SFT experiment in Section A.9.1 further supports this: post-trained models learn the binary-stability task faster than the base model, but formatting alone cannot explain the gap — after 100 SFT steps, the base model produces only legal answers yet still achieves lower accuracy. This suggests post-training instills some transferable physical representations, but these are not sufficient for zero-shot generalization without additional task-specific supervision.
> \
> \
> **Could the negative results be partly due to limitations of the Qwen model family?**
>
> Thank you for bringing the spurious rewards paper to our attention. We have repeated our experiments with other model families: we trained Gemma3-12B on the x-only side block task and Llama3.2-11B on the x-only top block task. We find that all models show the same failure to generalize (llama: https://postimg.cc/mtRNxms2, gemma: https://postimg.cc/MntpPCf5). Note that due to computational and time constraints of the rebuttal, we only have results for GRPO models; we will include SFT-trained models in the camera-ready version.
>
> We thank the reviewer for their comments and hope we have addressed their concerns. We hope we could convince the reviewer of the sufficient scope of our investigation. We believe our approach of using controlled experiments to pin-point where current post-training approaches fall short is a valuable contribution and we are confident it will be helpful in developing models and training paradigms with generalizable physical understanding.

---

> > ### Author Rebuttal · Reviewer_SyFm · 2026-04-01
> >
> > Thank you. I think the response resolved all my concerns.

---

> > > ### Author Response · Authors · 2026-04-02
> > >
> > > Dear Reviewer SyFm,
> > > \
> > > \
> > > Thank you for confirming that all of your concerns have been resolved. We deeply appreciate your recognition of the paper's significance and your assessment that the problem "is important and has meaningful potential impact."
> > > \
> > > \
> > > In your initial review you had noted that you would raise your score if more sound and in-depth insights were provided. We would like to kindly ask whether you might consider updating your final recommendation to reflect the current state of the paper.
> > > \
> > > \
> > > Thank you for your time and consideration — we greatly value your engagement with our work.

---

### Official Review · Reviewer_4HZ8 · 2026-02-27

**Soundness:** 3
**Presentation:** 4
**Significance:** 3
**Originality:** 3
**Overall Recommendation:** 5
**Confidence:** 4

**Summary:**

This paper tests whether Vision Language Models (VLMs) can learn generalizable intuitive physics by "interacting" with an environment, comparing 1-step reinforcement learning (GRPO) against supervised fine-tuning (SFT). The authors construct synthetic block-stacking tasks where models must either predict tower stability or output coordinates to place a block safely.

The findings are consistently negative. While both GRPO and SFT models easily master the training tasks, they fail to generalize to slightly modified tasks (e.g., moving a block from the top vs. the side) or to real-world images. Interestingly, linear probing shows that the base models actually encode the correct physical variables (like stability and x-offset) internally. However, the post-training methods fail to utilize this knowledge, instead teaching the models to exploit task-specific visual shortcuts.

**Compliance With Llm Reviewing Policy:**

Affirmed.

**Final Justification:**

The paper provides a clean, well-executed negative result demonstrating that RL post-training fails to instill generalizable physical understanding in VLMs. Weighing the paper's solid execution and timeliness against its limitations, I find it to be a valuable contribution. The authors' rebuttal successfully addressed my primary concerns—most notably by adding a multi-step RL experiment that strengthens their core claims and by correcting the SFT terminology. Because the rebuttal effectively reinforced the paper's soundness and clarified the remaining issues, I have confidently raised my score to a 5.

**Key Questions For Authors:**

1. Why frame this so heavily around "interaction"? A single-step prediction on a static image doesn't capture the temporal dynamics of play. Do you expect that a multi-step MDP formulation (where the model sees the tower fall and can try again) would actually bridge the generalization gap?
2. Because the camera angle is fixed, the task can be solved by simply measuring 2D pixel distances. Did you try randomizing the camera angle and distance during training? This might prevent 2D shortcut learning and force the model to rely on the 3D physical features you found in the decoding analysis.
3. For the GRPO reward, you used a continuous Gaussian based on distance. Did you try a sparse reward (e.g., +1 for stable, 0 for fall) to see if that forces the model to actually learn the physics of stability rather than just regressing your distance function?
4. In the generalization tests (e.g., moving from an x-only integer output to a binary stability output), how much of the failure is due to a lack of physical understanding versus the model simply failing to adapt to the new prompt format zero-shot?
5. Given that all experiments use QLoRA adapters, have you considered whether full fine-tuning might produce different generalization behavior? The PEFT setup limits the model to small perturbations around its pretrained weights, which may inherently constrain the kind of representations it can learn.

**Limitations:**

The authors are upfront about using a 1-step RL setup and relatively small-scale data and models. However, they understate the impact of the PEFT-only setup and the lack of visual diversity in training data, both of which are significant confounds for the paper's central claims.

**Strengths And Weaknesses:**

### Strengths:

* It's a clean, well-executed negative result. The community is currently highly optimistic about RL post-training for reasoning. Showing that it fails to instill actual physical understanding in VLMs provides a useful and timely reality check. The result directly contrasts with Chu et al. (2025), who found that RL generalizes better than SFT on arithmetic and navigation tasks — the present paper demonstrates this advantage does not hold for intuitive physics, which is a valuable domain-specific nuance.
* The decodability analysis is a great addition. Proving that the models possess the internal "competence" (features are linearly separable) but lack the "performance" out-of-distribution effectively highlights that the issue is stubborn shortcut learning, not just a blind vision encoder.
* Extensive ablations. The authors didn't just test one model; they evaluated different sizes (up to 32B), another RL algorithm (GSPO), varying training horizons (up to 48K steps), adapter ranks, and vision encoder freezing, which makes the negative results much more convincing.

### Weaknesses:

* Framing a 1-step RL update as "interaction" is a major stretch. The cognitive science motivation relies on continuous, temporally extended experimentation. Predicting a single integer and getting a scalar reward is essentially a contextual bandit problem on a static image, which completely removes the sequential dynamics of physical play. The paper should be more upfront about this mismatch rather than leaning so heavily on the embodied cognition framing.
* Relatedly, the paper frames SFT as "an analogue of offline reinforcement learning" (citing Levine et al., 2020). This is a misleading equivalence. SFT here is supervised learning on optimal demonstrations with a cross-entropy loss — it does not involve reward estimation, temporal credit assignment, or behavior policy correction, which are the defining features of offline RL. This framing overstates the conceptual gap between the two conditions and makes the comparison appear more principled than it is.
* The visual diversity of the synthetic training data is severely impoverished. The dataset uses a fixed camera angle, uniform blocks, and a gray background. It's practically guaranteed that a neural network will learn 2D pixel-level distance shortcuts here. The failure to transfer to real images might just be a standard CV domain gap issue rather than evidence about the limits of physical understanding.
* The GRPO reward function is an engineered Gaussian based on distance to the center. The model doesn't need to learn anything about gravity; it just has to regress this specific mathematical distance function.
* All experiments use QLoRA (Parameter-Efficient Fine-Tuning) rather than full fine-tuning. With only small low-rank adapters being updated (r=16) while the vast majority of model weights remain frozen, the model's capacity to learn new representations is fundamentally limited. It is possible that full fine-tuning — or at least updating a larger portion of the model — would yield different generalization patterns. This confounds the paper's conclusions about RL vs. SFT with conclusions about PEFT limitations.

---

> ### Author Rebuttal · Authors · 2026-03-31
>
> Dear reviewer 4HZ8,
>
> Thank you for your insightful review. We appreciate that you find our results "clean, well-executed" and "a useful and timely reality check", and that you like the decodability analysis and ablations. We address your remaining concerns below.
> \
> \
> **Framing 1-step RL as "interaction"**
>
> We agree our operationalization is more limited than the continuous physical play described in cognitive science, and have revised the introduction and limitations to be more explicit about this. That said, we believe our setup does involve meaningful, if limited, interaction: the model observes a physical scene, produces an action, and receives a reward depending on the physical outcome.
>
> In response to your and other reviewers' concerns, we added a multi-step version of the x-y side block task in which the model observes three frames depicting two successive actions before predicting the optimal final block placement. A model trained with GRPO on this multi-step task fails to generalize to the single-image version, and vice versa, despite sharing the same physical dynamics, visual characteristics, and reward functions (results: https://postimg.cc/P50r5v9c). This shows that even minimal changes to sequential structure can lead post-trained models to fail.
> \
> \
> **SFT as "analogue of offline RL"**
>
> We agree the framing was imprecise and have updated the paper to describe SFT as "akin to behavioral cloning," which better reflects the conceptual relationship. Thank you, we appreciate the push toward more careful framing.
> \
> \
> **Impoverished visual diversity**
>
> The limited visual diversity is intentional. By fixing camera angle and block sizes, we ensure pixel-action mappings are consistent across tasks. A model that learns the relationship between an action of "+100" and the corresponding pixel displacement could in principle transfer this mapping to the other tasks. We keep everything constant so that any failure to generalize cannot be attributed to perceptual confounds, but must reflect a deeper limitation. We argue that this makes our generalization failures more striking: we have removed every perceptual obstacle to transfer, yet models still fail. However we agree that diverse training suites with various tasks and more visual variance are an interesting future direction, and we have added this to the discussion. Regarding specific shortcuts: no obvious pixel-level shortcut presents itself for the x-only and x-y tasks, and attention maps did not reveal any consistent pattern.
> \
> \
> **Gaussian reward / sparse reward**
>
> For the binary stability task we already use a sparse reward (+1 correct, 0 incorrect, -1 illegal). This model also fails to generalize: it does not transfer to the other tasks, and while it improves somewhat over the base model on Lerer real images, it remains well below human performance and far below its accuracy on the fine-tuning task. For x-only and x-y tasks, the Gaussian reward keeps the reward structure consistent across conditions so that learned pixel-to-action mappings are in principle transferable — yet models still fail to generalize. We also attempted to train a model on a sparse version of the x-y task for the rebuttal, but the reward was too sparse for the model to learn the task at all.
> \
> \
> **QLoRA / PEFT limitations**
>
> Prior work has demonstrated that 4-bit quantization introduces negligible performance degradation compared to full-precision fine-tuning [1-3] and is optimal for zero-shot accuracy per bit [4]. We are therefore confident our results are not artifacts of quantization, and have added a discussion of this to the paper.
> \
> \
> **Formatting vs. physical understanding**
>
> We believe that formatting is not the primary driver of generalization failures. We have added a table of binary stability results with illegal responses excluded (https://postimg.cc/ZR6CVPWN) and a variant of Figure 2 also showing results for legal answers only (https://postimg.cc/nzr1n5mY/). Most models adhere to the required output format across tasks. A notable exception is the GRPO x-y side block model (57% illegal responses on binary stability): when restricting to its legal responses it achieves 0.74 accuracy on binary-stability top block — yet it still fails on the Lerer dataset.
> \
> \
> **Limitations section**
>
> We have expanded and made the limitations section more prominent, covering the single-step RL setup, model scale, PEFT constraints, visual diversity, and quantization, along with concrete directions for future work.
>
>
> We thank the reviewer for their interesting comments and hope our responses have addressed their concerns. We believe our approach of using controlled experiments to pin-point where current post-training approaches fall short is a valuable contribution and we hope it can serve as a reference toward developing models and training paradigms with generalizable physical understanding.
>
> Full reference list available here: https://postimg.cc/8jXkQqmx.

---

> > ### Author Rebuttal · Reviewer_4HZ8 · 2026-04-03
> >
> > The authors successfully addressed my primary concerns by providing the new multi-step RL experiments and clarifying the SFT terminology, so I am happy to raise my score to 5.

---

### Official Review · Reviewer_4pZg · 2026-03-12

**Soundness:** 2
**Presentation:** 2
**Significance:** 2
**Originality:** 2
**Overall Recommendation:** 3
**Confidence:** 4

**Summary:**

This paper investigates whether VLMs can acquire intuitive physical concepts through interaction with an environment. The authors train models on block-tower manipulation tasks using reinforcement learning based on GRPO, and compare the results with SFT. The tasks include predicting tower stability and generating actions to build stable towers by moving blocks in a simulated environment. The experimental setup includes several related synthetic block-tower datasets as well as an external dataset containing real images of block towers. The models are evaluated both on within-task performance and on their ability to generalize across tasks and datasets. The authors also conduct an activation decodability analysis to examine whether physical quantities such as tower stability or block displacement are encoded in the models’ internal representations.

**Compliance With Llm Reviewing Policy:**

Affirmed.

**Final Justification:**

Thank you for the additional discussion on potential solutions, which strengthens the practical relevance of the work. As I still have some concerns about the completeness of the current manuscript, I raise my score to the weak reject.

I have also read the other reviewers’ comments, and agree that the paper presents a clear and meaningful result demonstrating the failure of RL to generalizable physical understanding. If they feel positive about the manuscript, I would be supportive of accepting this paper.

**Key Questions For Authors:**

1. Figure 1 is somewhat confusing. What exactly does the highlighted region indicate? In addition, for the x-y task, what does “moved up” mean in practice? Is the model allowed to pick up a block and place it in the table, or must the moved block always be in the air?

2. The experiments mainly focus on relatively simple tower manipulation tasks. Do the authors expect the conclusions to extend to more complex physical environments involving multi-step interactions or richer object dynamics? Have the authors considered evaluating the models on other intuitive physics benchmarks beyond block towers?

3. The fine-tuning experiments are conducted only on Qwen3-VL. I wonder whether the same conclusion would hold for other VLM families, such as InternVL.

4. In Figure 2, the performance of both SFT and GRPO on the x-only side block and x-y side block tasks appears to decrease as training proceeds. What is the reason for this behavior?

**Limitations:**

yes

**Strengths And Weaknesses:**

Strengths

- One strength of this paper is that it explores an interesting question, i.e., whether interaction-based learning can improve the acquisition of intuitive physics in vision-language models. The motivation is well grounded in insights from cognitive science, which suggest that humans develop physical intuitions through active interaction with their environment.

- The authors construct several related datasets and tasks, including binary stability judgment, x-offset correction, and x-y action prediction, and evaluate the models across these settings to study generalization. The inclusion of real-image evaluation further strengthens the empirical analysis by testing whether the models can transfer beyond synthetic environments.

- Another positive aspect is the decodability analysis, which helps reveal whether relevant physical information is already represented in the model activations. This provides a useful way to distinguish between model competence and model performance.

Weaknesses

- Although the paper aims to investigate whether interaction improves physical reasoning, the notion of interaction used here is relatively limited. The experiments only involve simple tasks such as block manipulation, which may not be sufficient to test the broader hypothesis that interaction leads to more general and transferable physical understanding. The tasks and datasets are also relatively constrained, as most of them are variations of the same tower configuration problem. This limits the diversity of physical scenarios being evaluated, and therefore the conclusions about general intuitive physics may not easily extend to more complex physical reasoning tasks.

- The results show that both GRPO and SFT enable the models to achieve near-ceiling performance on the tasks they are trained on. However, neither training strategy leads to robust generalization across related tasks or datasets. The authors conclude that current post-training methods may primarily encourage shortcut learning rather than genuine acquisition of physical intuition.

- The experimental section also feels incomplete. While the paper provides several figures showing the reward values achieved by the models on different tasks, it currently lacks tables that would allow more direct quantitive comparisons. In addition, intuitive physics is often evaluated using more standard metrics, such as accuracy on stability prediction (e.g., whether a tower will fall). Such metrics are not sufficiently emphasized in the experimental analysis.

- The paper suggests that fine-tuning through interaction in physical environments may improve intuitive physics learning. This is an interesting claim, but it is still not entirely clear how the findings could further inform improvements in training strategies.

---

> ### Author Rebuttal · Authors · 2026-03-31
>
> Dear reviewer 4pZg,
>
> Thank you for your thorough review. We appreciate that you agree we address an "interesting question" and that our motivation is "well grounded in insights from cognitive science", and that you appreciate the real-image evaluation and decodability analysis. We address your concerns below.
> \
> \
> **Limited notion of interaction / constrained tasks**
>
> The simplicity of our tasks is intentional. Following a long tradition in developmental psychology [1] and machine learning [2, 3], we use constrained, controlled stimuli to precisely isolate where generalization fails. Since pixel-action mappings are consistent between conditions and visual stimuli share the same characteristics, any failure to generalize cannot be attributed to perceptual confounds. The key finding is that even in this maximally favorable setting, models fail to generalize — directly challenging the belief that RL post-training yields transferable representations [4].
> In response to your and other reviewers' concerns, we have added a multi-step experiment in which models observe three frames depicting two successive actions before predicting the optimal final placement. A model trained on this triplet task learns to perform it well, but fails to generalize to the single-image version, and vice versa, despite sharing the same dynamics, visual characteristics, and reward function (results: https://postimg.cc/P50r5v9c).
> \
> \
> **Summary of findings**
>
> We agree with the reviewer's characterization. The decodability analysis further supports this: the relevant physical variables are represented within the models but not leveraged for generalizable reasoning, suggesting post-training drives shortcut learning rather than genuine physical intuition.
> \
> \
> **Missing tables / relation to accuracy**
>
> We have added a combined main comparison table to page 6.
>
> For binary stability tasks, our reward is equivalent to accuracy for legal responses (models receive rewards of 1 correct, 0 incorrect, -1 illegal). We added a table to Appendix A.2 for results with only legal responses. All models adhere to the required format except for GRPO x-y side block (57% illegal responses), which achieves 0.74 accuracy on legal responses but still fails on the Lerer dataset (table: https://postimg.cc/ZR6CVPWN; see also: https://postimg.cc/nzr1n5mY for Fig. 2 with only legal answers).
> \
> \
> **Implications for training strategies**
>
> Our take-away is not that interaction improves intuitive physics learning generally. Rather, both GRPO and SFT improve within-task performance but neither yields generalizable understanding. This implies that scaling RL post-training on a single task family is unlikely to produce robust physical intuition. Larger scale training on various environments and training objectives that encourage models to ground predictions in physical variables rather than surface shortcuts could improve generalization. We have added these suggestions for future work in the discussion.
> \
> \
> **Figure 1 clarity / x-y task**
>
> We have simplified Figure 1 and added a sentence clarifying that heatmaps visualize the reward functions (updated figure: https://postimg.cc/VNjbJNGn). For the x-y task, the model returns two integers moving the block left/right and up/down. Illegal positions (e.g., inside the table) are penalized, so models quickly learn to respect physical constraints.
> \
> \
> **Extension to more complex environments**
>
> We address this in the multi-step experiment above. If models cannot bridge the gap between a single-frame and three-frame version of the same task, it is difficult to be optimistic about generalization to richer environments. The bottleneck is not task complexity per se: models likely rely on task-specific shortcuts rather than transferable representations. Evaluation on broader intuitive physics benchmarks is a natural future direction, but we suspect the same pattern would emerge.
> \
> \
> **Other VLM families**
>
> We trained Gemma3-12B on x-only side block and Llama3.2-11B on x-only top block with GRPO. Both show the same failure to generalize as the Qwen models (llama: https://postimg.cc/mtRNxms2, gemma: https://postimg.cc/MntpPCf5). SFT-trained versions will be included in the camera-ready.
> \
> \
> **Decreasing performance in Figure 2**
>
> The diagonal shows within-task performance, which increases or plateaus throughout training. The decreases occur in off-diagonal cells which show transfer tasks. As models learn their training task, they actively move away from solutions that would transfer, underscoring how brittle their learned ability is.
> \
> \
> We thank the reviewer for their thoughtful engagement and hope our responses have addressed their concerns. We agree that building models with robust physical intuition is an important goal and believe our paper makes a meaningful contribution toward it by carefully documenting where current post-training approaches fall short under controlled conditions.
>
> Full reference list: https://postimg.cc/nsjJJrh2.

---

> > ### Author Rebuttal · Reviewer_4pZg · 2026-04-03
> >
> > Thank you for the response. It addresses most of my concerns. I agree with your analysis that the model may encode relevant physical variables but rely on shortcut learning rather than generalizable physical reasoning.
> > To strengthen the practical impact of this work, I would strongly encourage to propose a concrete solution or direction for enabling transferable, generalizable reasoning rather than only diagnosing the limitation. Addressing how to move beyond shortcut learning toward would significantly enhance the method’s utility and contribution to the field.

---

> > > ### Author Response · Authors · 2026-04-04
> > >
> > > We thank the reviewer for their continued engagement and are glad our responses addressed most of their concerns. We agree that pointing toward solutions is important, and would have elaborated more on this in our initial rebuttal had character limits allowed. We have added the following section to the discussion to clearly highlight potential solutions:
> > >
> > > >Our findings point toward a potential direction for overcoming models' inability to generalize: training models on more diverse training distributions including multiple tasks. We find that simply scaling RL post-training on a single task is unlikely to produce robust physical intuition — our results show that longer training horizons only lead to further overfitting rather than broader generalization (see Section A.9.2 on longer training horizons in the Appendix). However, our results on multi-task training point toward a path forward: when models are trained with GRPO on two tasks sequentially, they retain the ability to perform both (see Section A.9.4 on joint training in the Appendix). This is a preliminary but encouraging signal that multi-task training regimes can overcome some of the brittleness we observe in single-task post-training.
> > > \
> > > \
> > > Additionally, the decodability analysis shows that the relevant physical variables are already represented in the models but not recruited for generalization. Taken together, we think that future work should explore more diverse training distributions and curriculum-based approaches that gradually introduce more complex physical scenarios, as well as auxiliary objectives that encourage models to ground predictions in physical variables rather than surface shortcuts.
> > >
> > > \
> > > We hope the reviewer will consider our careful documentation of current post-training approaches' limitations — and the concrete potential solutions we have now outlined — as a strong foundation for the follow-up work they are rightly encouraging, and that taken together with our previous responses, this might warrant a reconsideration of their score.

---

### Official Review · Reviewer_Hn4b · 2026-03-21

**Soundness:** 2
**Presentation:** 2
**Significance:** 2
**Originality:** 2
**Overall Recommendation:** 2
**Confidence:** 4

**Summary:**

The paper aims to address the limitations of vision-language models in understanding intuitive physics in real-world settings. The authors show that acquiring such intuitive physical understanding remains challenging, even when the model interacts with the environment. Overall, the paper attempts to tackle an interesting problem; however, in its current form, it appears premature. The presentation suffers from multiple formatting issues, missing comparison tables, limited discussion of related work, outdated references, and an unclear problem definition.

**Compliance With Llm Reviewing Policy:**

Affirmed.

**Final Justification:**

I still have two unresolved concerns. First, the paper repeatedly describes the method as learning from interaction, yet the clarification that no robot arm or environmental interaction is involved makes this characterization unclear to me. If the setup is limited to asking questions such as whether a stack will fall, I do not see why this should be framed as interaction-based learning.

My main concern, however, is the second point. I am not convinced that the distinction the authors draw between intuitive and formal physics is sufficiently different from prior VLM physics evaluation benchmarks. Those benchmarks also ask similar kinds of questions and, as the authors acknowledge, can already be used to evaluate intuitive physics. Because this issue is central to the paper’s claimed novelty, I do not think my concern has been resolved, and I am maintaining my score.

**Key Questions For Authors:**

See Weakness.

**Limitations:**

No. The paper does not include a limitations section. Although the authors do provide a very brief impact statement without specifying any of the limitations of the proposed approach. Adding a brief limitations section would strengthen the paper and help clarify the method’s assumptions, boundaries, and potential failure cases.

**Strengths And Weaknesses:**

**Strength**

The attempt to improve a VLM’s ability to understand intuitive physics through interaction is novel. If successful, such a capability could be highly valuable to the community.

**Weakness**
1. It is unclear whether the paper is discussing vision-language models (VLMs) or vision-language-action models (VLAs) in the context of robotics tasks. These are different model classes, and this distinction matters, especially since the experiments appear to involve manipulation and action prediction, which are more closely aligned with VLAs.
2. In related works, when authors highlight VLMs ability to distinguish between multiple objects they cite relatively old research. The reviewer believes that recent VLMs are much better in those tasks (line 108-109). Given the rapid progress in this area, the paper would benefit from engaging more thoroughly with recent work, particularly newer VLMs and VLAs relevant to robotics.
3. It would also benefit the readers from understanding how intuitive physics is different from normal physics and how this paper is positioned w.r.t previous research in understanding physical world understanding of VLMs.
4. More broadly, the related work section focuses heavily on describing the method setup rather than positioning the paper against prior work that studies intuitive physics, embodied reasoning, or action-oriented vision-language systems. A clearer discussion of closely related recent directions would strengthen the motivation.
5. The experiments use the 8B Qwen3-VL model with 4-bit quantization. It is unclear how much this choice affects the results. Since quantization can degrade model capabilities, it would be helpful to know whether similar trends hold without quantization, or at least to include discussion of this tradeoff.
6. It is also unclear whether the authors evaluated the robustness of their findings across different prompting strategies. Since prompt design can strongly influence VLM performance, some analysis of prompt sensitivity would make the conclusions more convincing.
7. The reward functions appear highly task-specific and are not defined in a consistent manner across experiments. The paper would benefit from stronger justification for these design choices. In addition, some rewards seem quite sparse, especially for tasks such as binary stability prediction, which may make learning unnecessarily difficult.
8. The empirical study is limited to the Qwen family. As a result, the claim that VLMs perform poorly on these tasks may be too broad unless the analysis is extended to other model families as well.
9. It is unclear whether the authors compared against other reinforcement learning methods, such as PPO, or only considered GRPO. Including such comparisons, or explaining why they were omitted, would strengthen the experimental section.
10. The paper does not clearly report the amount of training data used for GRPO. Since data scale is important for RL-based fine-tuning, this detail is necessary for interpreting the results. Poor performance may partly reflect limited training data rather than the inability of the model class itself.
11. The presentation also needs improvement. There are several formatting issues, including large white spaces on pages 1, 3, and 8. In addition, the main comparison table appears to be missing from the main paper and is instead placed in Appendix A.2. The experimental results are also not discussed in enough detail, making it difficult to draw clear conclusions from the reported numbers alone.

---

> ### Author Rebuttal · Authors · 2026-03-30
>
> Dear reviewer Hn4b,
>
> Thank you for your in-depth review. We appreciate your assessment of our paper as tackling "an interesting problem" with a "novel" approach. We understand that you retain multiple concerns about our paper and we address them individually below:
> \
> \
> **VLMs vs. VLAs**
>
> Our paper focuses on VLMs, however our setup shares key features with VLA settings: models receive visual observations and produce actions that affect the environment. We believe our findings are relevant to the broader question of whether interaction-based training can instill generalizable physical understanding.
> \
> \
> **Recent VLMs are better at multi-object tasks**
>
> We agree, and have updated our related work to reflect recent progress on multi-object discrimination and counting [1-9].
> \
> \
> **Intuitive physics vs. formal physics**
>
> Intuitive physics refers to the fast, implicit understanding of physical processes that humans deploy rapidly and without deliberation — it is probabilistic, approximate, and grounded in perception and action. Unlike formal physics, it does not rely on explicit mathematical laws. Its canonical components include object permanence, continuity, solidity, and support [10-12]. Our work studies these intuitions in VLMs rather than formal physics [13-17]. We have also highlighted this distinction in the introduction.
> \
> \
> **Related work positioning**
>
> We have substantially revised the related work section. Prior work documents broad VLM failures on physical reasoning benchmarks [18-19], and probing analysis shows vision encoders capture physical plausibility cues even when behavior fails [18], directly motivating our decodability analysis. Recent work on stability tasks shows SFT and RL can improve task performance but does not test cross-task generalization [19]. World modeling benchmarks suggest VLMs lack robust representations of conservation and causality [20-21]. Complementarily, V-JEPA [22] shows intuitive physics can emerge from video pretraining, achieving 98% zero-shot accuracy on IntPhys [23].
> \
> \
> **4-bit quantization**
>
> Prior work shows 4-bit quantization introduces negligible performance degradation [24-26] and is optimal for zero-shot accuracy per bit [27]. We have added a discussion of this tradeoff to the limitations section.
> \
> \
> **Prompt sensitivity**
>
> We ran a prompt sensitivity analysis testing four prompt variations on 1,000 images per task (prompts: https://postimg.cc/k6rbMVdZ; results: https://postimg.cc/hz37dRpv). While we find some performance differences across prompts, the overall pattern is consistent: fine-tuned models perform well on their training task but fail to generalize. We have added these results to the Appendix.
> \
> \
> **Reward function design**
>
> Our reward functions are consistent across conditions: models trained on x-only top block, x-only side block, and x-y side block tasks use the same Gaussian reward functions, meaning they should in principle be able to generalize. You are correct that the binary stability task uses a sparse reward (+1 correct, 0 incorrect, -1 illegal), however we find all models, regardless of whether their reward function was sparse or Gaussian, fail to generalize reliably.
> \
> \
> **Other model families**
>
> We trained Gemma3-12B on x-only side block and Llama3.2-11B on x-only top block with GRPO. Both show the same failure to generalize as the Qwen models (llama: https://postimg.cc/mtRNxms2, gemma: https://postimg.cc/MntpPCf5). SFT-trained versions will be included in the camera-ready.
> \
> \
> **Other RL methods**
>
> Please see Section 4.5.2. We train with GSPO on the x-only top block task for both 8B and 32B models, and find the same pattern of results. We focus on GRPO and GSPO rather than PPO as these are more effective and require no value model [28].
> \
> \
> **Training data scale**
>
> We train on 10,000 images per task combination. Models are evaluated on a separate set of 10,000 images per task. We have added this to the methods section.
> \
> \
> **Presentation**
>
> We have removed white spaces, added a main comparison table to page 6, redesigned Figure 1 (https://postimg.cc/3yxw63Hx), and substantially revised the results section to explicitly connect each finding to our core claims rather than reporting numbers in isolation.
> \
> \
> **Limitations section**
>
> We have made the limitations section more prominent and expanded it to cover quantization, model scale, data diversity, and the single-step nature of our interaction setting, along with directions for future work.
> \
> \
> We thank the reviewer for their thorough engagement with our work. We believe our paper makes a meaningful contribution by carefully documenting where current post-training approaches fall short under controlled conditions designed to give models every opportunity to succeed — and we hope it serves as a useful reference point toward training paradigms that produce models with generalizable physical understanding.
>
> Full reference list available here: https://postimg.cc/ZWb2TRVv.

---

> > ### Author Rebuttal · Reviewer_Hn4b · 2026-04-02
> >
> > I thank the authors for their detailed rebuttal. While the authors have provided more details/analysis, and  have promised to improve the paper’s clarity, I still hold some concerns regarding the conceptual framing and the alignment between the models used and the tasks evaluated.
> >
> > 1. Framing of Intuitive vs. Formal Physics: The distinction between "intuitive" and "formal" physics still remains confounded. While the authors define intuitive physics as an implicit, perceptual understanding, the experimental setup, which requires precise robot arm manipulation, inherently demands a more formal or procedural physical understanding.
> >
> >     If the objective is to evaluate intuitive physics, the evaluation should focus on the model’s ability to infer physical properties, affordances, and constraints (e.g., "Will this stack fall?" or "Is this object reachable?"). Moving a block with a robot arm introduces complexities that shift the task toward control and formal dynamics. I recommend the authors look into some recent benchmarks of VLMs for physical understanding, such as [1] and [2].
> >
> > 2. Model-Task Alignment (VLM vs. VLA): There is a fundamental mismatch between the model class being tested and the task being performed. VLMs are traditionally designed for vision-language reasoning (VQA), not direct environmental control. Testing a VLM’s "physics understanding" via robot manipulation is an uneven evaluation unless compared against models actually architected for action.
> >
> >     If the authors want to focus on "interaction" and "manipulation" in the paper, they should utilize a Vision-Language-Action (VLA) model. Models like VLA-0 [3] would be a more appropriate choice as it is specifically designed to leverage VLM backbones for manipulation.
> >
> > While the additional analyses were provided, the core methodological gap still remains. I believe a significant restructuring of the evaluation protocol is necessary to clearly differentiate this work from existing benchmarks and to justify the choice of model for the stated research direction. With that, as it stands now, I would like to maintain my score.
> >
> > [1] Gundawar, Atharva, Som Sagar, and Ransalu Senanayake. "PAC Bench: Do Foundation Models Understand Prerequisites for Executing Manipulation Policies?." arXiv preprint arXiv:2506.23725 (2025).
> >
> > [2] Chow, Wei, et al. "Physbench: Benchmarking and enhancing vision-language models for physical world understanding." arXiv preprint arXiv:2501.16411 (2025).
> >
> > [3] Goyal, Ankit, et al. "Vla-0: Building state-of-the-art vlas with zero modification." arXiv preprint arXiv:2510.13054 (2025).

---

> > > ### Author Response · Authors · 2026-04-02
> > >
> > > Dear Reviewer Hn4b,
> > >
> > > Thank you for your continued engagement with our work. We appreciate the opportunity to address your remaining concerns, and we hope the clarifications below can help resolve what we believe are a few persistent misunderstandings about our setup.
> > > \
> > > \
> > > **On the VLM vs. VLA distinction:**
> > >
> > > We would like to reiterate that our paper studies vision-language models (VLMs) exclusively — we do not train or evaluate a single VLA. We understand this distinction may not have been sufficiently prominent in the original submission, and we have revised the introduction to make this unambiguous. We believe our work is precisely about whether VLMs, post-trained via RL, can acquire generalizable physical understanding — which is a distinct and important research question from evaluating VLAs on manipulation tasks.
> > > \
> > > \
> > > **On robot arm manipulation:**
> > >
> > > We want to respectfully but clearly address the concern that our setup "requires precise robot arm manipulation" and involves "moving a block with a robot arm." There is no robot arm in our paper. The models output at most two integers representing a displacement in a simulated environment; no physical robot is involved at any point. We are uncertain how this impression arose, as the manuscript contains no mention of a robot, and we want to ensure this does not factor into the evaluation of our work.
> > > \
> > > \
> > > **On the intuitive vs. formal physics framing:**
> > >
> > > Our research question is whether models can learn intuitive physics from being trained interactively on physical tasks — including tasks that may benefit from formal physical understanding. Specifically, we ask whether interactive training on physical tasks confers an advantage in learning general intuitive physical understanding. Your suggestion that intuitive physics evaluation should focus on inferring physical properties and constraints — such as predicting whether a stack will fall — is well-taken, and it accurately describes what we do. Our evaluation includes precisely these tasks (including tower stability prediction and decoding physical properties from the models' representations), and we test whether models trained on one physical task can generalize to related ones. Crucially, as you suggest, we evaluate all models on the canonical tower stability task, asking them "Will this stack fall?". Our central finding is that even under maximally favorable conditions — matched visual statistics, reward functions, and physical parameters — interaction-based post-training fails to produce transferable intuitive physical understanding. We believe this is a meaningful and timely result, given recent claims that RL post-training yields models that generalize better than models post-trained with SFT [1].
> > > \
> > > \
> > > We hope these clarifications address the remaining concerns, and we sincerely ask that you reconsider your score in light of them. We are happy to provide any further clarification.
> > > \
> > > \
> > > [1] Chu et al., "SFT Memorizes, RL Generalizes," arXiv:2501.17161 (2025).

---

### Decision · Program_Chairs · 2026-04-30

**Decision:**

Accept (regular)

**Comment:**

This article investigates a timely question: Can vision language models (VLMs) learn generalizable physical intuitions by interacting with a simulated environment, via RL training? While the task-specific accuracy increases through training (as expected), the article reports that generalization to similar tasks is severely limited: models learn not generalizable physical principles, but shortcuts.

During the rebuttal discussion, the authors were able to convince 3 out of 4 reviewers (reviewer 4pZg has score weak reject but also mentioned that, after reading other reviewers' reviews, they see the value of the article and would not be opposed to publication at ICML). Reviewer Hn4b is favoring rejection; they worked with the authors to improve clarity. Reviewer Hn4b is concerned that vision-language-action models (VLAs) might be a better choice for interaction-based experiments. The AC agrees that this is a valuable suggestion for an alternative approach; however, investigating a "simulated interaction" (as the authors do in a virtual environment) is a valuable contribution, too. As such, the AC is supportive of publication at ICML and would encourage the authors to make it crystal clear in their manuscript (including the abstract, and possibly the title) that they are working with a _simulated_ (not real) environment.